# Photoreceptor degeneration has heterogeneous effects on functional retinal ganglion cell types

Nadine Dyszkant[1,2,3], Jonathan Oesterle[1,2], Yongrong Qiu[1,2,4,5], Merle Harrer[1,2], Timm Schubert[1,2], Dominic Gonschorek[1,2] and Thomas Euler[1,2,3]

[1] *Institute for Ophthalmic Research, University of Tübingen, Tübingen, Germany*
[2] *Werner Reichardt Centre for Integrative Neuroscience, University of Tübingen, Tübingen, Germany*
[3] *Graduate Training Centre of Neuroscience (GTC), University of Tübingen, Tübingen, Germany*
[4] *Department of Ophthalmology, Byers Eye Institute, Stanford University School of Medicine, Stanford, CA, USA*
[5] *Stanford Bio-X and Wu Tsai Neurosciences Institute, Stanford University, Stanford, CA, USA*

Handling Editors: Nathan Schoppa & Jonathan Demb

The peer review history is available in the Supporting Information section of this article (https://doi.org/10.1113/JP287643#support-information-section).

**Abstract figure legend** Photoreceptor loss has heterogeneous effects on functional retinal ganglion cell (RGC) types. *A*, illustrations of cross-sections of *rd10* retinae at different time points during degeneration. Left: P30, rod degeneration has commenced, cones are still unaffected. Centre: P45, rod degeneration is complete and secondary cone degeneration has commenced. Right: P90, advanced degeneration, remaining cones are deformed. All: rod and cone photoreceptors are highlighted in khaki and yellow, respectively. RGCs are coloured according to their functional super-group, 'On'-RGCs in light blue, 'Off'-RGCs in dark blue and 'On–Off'-RGCs in pink. *B*, relative abundance of functional RGC super-groups coloured according to their super-group.

**Nadine Dyszkant** earned her bachelor's degree in Biosciences from Westfälische-Wilhelms-University in Münster (Germany) in 2018, followed by a master's degree in biology from Carl-von-Ossietzky University in Oldenburg (Germany) in 2021. Her longstanding interest in neuroscience – particularly in the mechanisms underlying sensory systems – led her to the University of Tübingen, where she is currently pursuing a PhD in the laboratory of Prof. Thomas Euler at the Institute for Ophthalmic Research and Centre for Integrative Neuroscience. Her research focuses on retinal function and the progression of retinal diseases, including conditions such as retinitis pigmentosa.

This article was first published as a preprint. Dyszkant N, Oesterle J, Qiu Y, Harrer M, Schubert T, Gonschorek D, & Euler T. 2024. Photoreceptor degeneration has heterogeneous effects on functional retinal ganglion cell types. bioRxiv. https://doi.org/10.1101/2024.09.06.610955

**Abstract** Retinitis pigmentosa is a hereditary disease-causing progressive degeneration of rod and cone photoreceptors, with no effective therapies. Using *rd10* mice, which mirror the human condition, we examined its disease progression. Rods deteriorate by postnatal day (P) 45, followed by cone degeneration, with most photoreceptors lost by P180. Until then, retinal ganglion cells (RGCs) remain light-responsive under photopic conditions, despite extensive outer retinal remodelling. However, it is still unknown if distinct functional RGC types alter their activity or are even lost during disease progression. Here, we asked if and how the response diversity of functional RGC types changes with *rd10* disease progression. At P30, we identified all functional wild-type RGC types also in *rd10* retinae, suggesting that at this early degenerative stage, the full breadth of retinal output is still present. Remarkably, we found that the fractions of functional types changed throughout progressing degeneration between *rd10* and wild-type: responses of RGCs with 'Off'-components ('Off' and 'On–Off' RGCs) were more vulnerable than 'On'-cells, with 'Fast On' types being the most resilient. Notably, direction-selective RGCs appeared to be more vulnerable than orientation-selective RGCs. In summary, we found differences in resilience of response types (from resilient to vulnerable): 'Uncertain' > 'Fast On' > 'Slow On' > 'On–Off' > 'Off'. Taken together, our results suggest that *rd10* photoreceptor degeneration has heterogeneous effects on functional RGC types, with distinct sets of types losing their characteristic light responses earlier than others. This differential susceptibility of RGC circuits may be of relevance for future neuroprotective therapeutic strategies.

(Received 11 September 2024; accepted after revision 17 January 2025; first published online 8 February 2025)
**Corresponding author** D. Gonschorek and T. Euler: Institute for Ophthalmic Research, University of Tübingen, Germany. Email: dominic.gonschorek@cin.uni-tuebingen.de and thomas.euler@cin.uni-tuebingen.de

**Key points**

- Retinitis pigmentosa is a hereditary disease causing progressive degeneration of rod and cone photoreceptors, with no effective therapies; it can be investigated using mutant mouse models, like *rd10*, that mirror the human condition.
- Recent studies found that retinal ganglion cells (RGCs) in *rd10* remain light-responsive, despite extensive loss of photoreceptors and outer retinal remodelling; however, specific RGC types still may change or even lose their functional response profile during early degeneration.
- Using two-photon calcium imaging, we assessed if and how the light-evoked activity of RGCs, and, hence, the retinal output to the brain, differs during the disease progression in *rd10* compared to wild-type mice.
- We found differences in the resilience of functional RGC types: generally, 'On'-types were more resilient than 'On–Off' or especially 'Off' types.
- Our data suggest that interventions may be more effective in the 'On' pathways, which turned out to be more resilient in *rd10*.

## Introduction

In an ageing population, an increasing number of patients suffer from blindness due to degenerative retinal diseases. For example, retinitis pigmentosa (RP), a hereditary degeneration of rod and cone photoreceptors, is highly heterogeneous with over 80 gene loci affected (reviewed in Ali et al., 2017; Humphries et al., 1992; Loukovitis et al., 2021; Phelan & Bok, 2000). Consequentially, the available model systems for studying retinal degeneration (*rd*) are similarly diverse (reviewed in Agurtzane Rivas & Vecino, 2009; Chader, 2002; Farber et al., 1994; Fauser et al., 2002).

A well-established model for studying the consequences of such a photoreceptor degeneration on retinal circuits is the *rd10* mouse (Chang et al., 2002). In this mutant mouse strain, a missense mutation in exon 13 of the $\beta$-subunit of the rod phosphodiesterase gene (Pde6$\beta$) triggers rod photoreceptor (rod) degeneration, which is followed by a secondary degeneration of the genetically intact cone photoreceptors (cones), eventually leading to functional blindness. Because genetic factors and likely the disease mechanisms are comparable to RP in human patients (reviewed in Hamel, 2014), *rd10* mice are considered a suitable model for this type of hereditary retinal

dystrophy. In *rd10* mice, first signs of rod deterioration can be detected around postnatal day (P) 16, with rod loss peaking around P20, and being completed by P45 (Barhoum et al., 2008; Puthussery et al., 2009; Samardzija et al., 2012). After ~6 months of age (~ P180), the animals possess virtually no cones; thus, no photoreceptors are left at this stage of degeneration in the outer retina (for overview, see Fig. 1). Typically, the degeneration starts in the mid-periphery, from where it progresses across the retina (Barhoum et al., 2008; Chang et al., 2007; Gargini et al., 2007), adding a spatial axis to the degeneration.

Due to the photoreceptor loss, the outer retina of *rd10* mice is substantially reorganized (Barhoum et al., 2008; Gargini et al., 2007; Phillips et al., 2010; Puthussery et al., 2009; Strettoi et al., 2003), resulting in new activity patterns, such as spontaneous oscillations (Biswas et al., 2014; Goo et al., 2011; Haselier et al., 2017; Stasheff, 2008; Stasheff et al., 2011; Toychiev et al., 2013). Degeneration of cells in the outer retina is followed by structural remodelling or rewiring of bipolar cells (BCs) (reviewed in Hoon et al., 2014; Marc et al., 2003), while their postsynaptic circuits are thought to remain intact. For example, retinal ganglion cells (RGCs) seem to keep their normal dendritic morphology and projection patterns up to at least 9 weeks (~ P60), when most photoreceptors have perished (Mazzoni et al., 2008). Accordingly, recent studies in different RP mouse models, including *rd10*, found that cones remain light-responsive and RGCs display light-evoked activity under photopic conditions until at least 2–3 months (~ P60) of degeneration (Ellis et al., 2023; Scalabrino et al., 2022).

However, even if overall light-driven activity appears similar to that in healthy wild-type controls, specific RGC types still may change or even lose their functional response profile during the early phase of degeneration.

This notion is supported by work showing that (partial) photoreceptor loss – experimentally initiated (Care et al., 2019; Lee et al., 2022) or disease-related (Ellis et al., 2023; Scalabrino et al., 2022) – affects RGC receptive fields (RFs). It is unknown, though, how the more than 40 different types of mouse RGCs (Baden et al., 2016; Bae et al., 2018; Goetz et al., 2022) are impacted by outer retinal degeneration. For an optic nerve crush (ONC) paradigm, it was recently shown that RGC types are differentially affected: using transcriptomic data to investigate the survival of mouse RGC types after ONC (Tran et al., 2019) showed that some types were more resilient and survived longer, whereas others were more vulnerable and degenerated soon after the damage. This suggests that RGC types can differ in their susceptibility to an insult.

In this study, we addressed if photoreceptor degeneration can lead to differential changes of low photopic light response properties in RGCs over disease progression. To this end, we used two-photon (2P) $Ca^{2+}$ imaging to systematically record light-evoked RGC responses in the explanted *rd10* retina and age-matched wild-type animals from P30 to P180. We found that in *rd10* a substantial number of RGCs remained light-responsive at P90 but almost none at P180, in line with earlier work (Ellis et al., 2023). Notably, the RGC soma density remained virtually the same as in wild-type across the investigated age span. However, other than expected from earlier studies, we detected the first functional effects of photoreceptor degeneration on (low photopic) RGC light responses already at P30, when 'Off'-responding RGC types were less frequent in *rd10* than 'On'-responding ones. Remarkably, other functional RGC types were affected later (e.g. 'Fast On' RGCs decreased after P45), whereas the fraction of cells grouped into the 'Uncertain' RGCs remained largely unchanged during the investigated period.

Together, our data suggest that specific functional RGC types are more vulnerable (or resilient) than others to an insult such as photoreceptor loss. This differential susceptibility of RGC circuits may be of relevance for future neuroprotective therapeutic strategies.

## Methods

### Ethical approval

All animal experiments were conducted at the University of Tübingen. They were approved by the institutional animal welfare committee of the University of Tübingen and performed according to the laws governing animal experimentation issued by the German Government (Regierungspräsidium Tübingen, §4 animal protocol: CIN03/21M, section 1.3.6). We also confirm that we understand *The Journal of Physiology*'s ethical principles

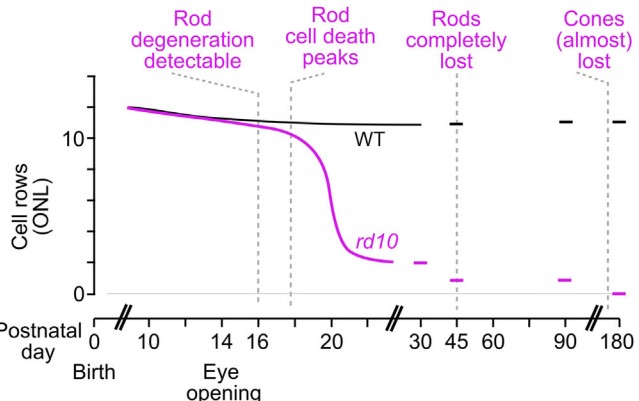

**Figure 1. Progression of photoreceptor degeneration in the *rd10* retina**
Rows of cells in the outer nuclear layer (ONL) for wild-type (WT, black) and *rd10* (magenta) retinae over time (for details and references, see text).

and we comply with the checklists given to authors (Grundy, 2015; O'Halloran, 2024).

## Animals and tissue preparation

We used retinae from C57BL/6J (wild-type, JAX 000664) and Pde6brd10 (*rd10*, JAX 004297) mice of either sex at postnatal days (P) 30, 45, 90 and 180 ($\pm$3 days). For recording experiments, we used $N = 23$ wild-type animals ($N = 36$ eyes) and $N = 25$ *rd10* animals ($N = 37$ eyes). For our immunohistochemistry stainings we used $N = 10$ wild-type animals ($N = 20$ eyes) and $N = 13$ *rd10* animals ($N = 26$ eyes). All animals were housed in the local animal facility under the standard 12 h/12 h day/night cycle at 22°C and a humidity of 55%. All animals had *ad libitum* access to food and water.

The following procedures were carried out under very dim red (> 650 nm) light. Before each experiment, the animal was dark-adapted for > 1 h, then anaesthetised with 5% gaseous isoflurane (CP-Pharma, Burgdorf, Germany) led into their cage, and sacrificed by cervical dislocation. Immediately after, the eyes were enucleated with a dorsal cut as orientation landmark and hemisected in carboxygenated (95% $O_2$, 5% $CO_2$) artificial cerebrospinal fluid (ACSF) containing (in mM): 125 NaCl, 2.5 KCl, 2 $CaCl_2$, 1 $MgCl_2$, 1.25 $NaH_2PO_4$, 26 $NaHCO_3$, 20 glucose, and 0.5 L-glutamine at pH 7.4. Sulforhodamine-101 (SR101, $\sim 0.1\mu$M; Thermo Fisher Scientific, Waltham, MA, USA) was added to the ACSF to visualise blood vessels for orientation and damaged cells in the red fluorescence channel of the microscope (Euler et al., 2009). The recording chamber was constantly perfused with carboxygenated ACSF at 4 ml/min and kept at ~36°C during the experiment.

After dissection, retinae were flat-mounted on thin ceramic discs with the ganglion cell layer (GCL) facing up (Anodisc #13, 0.1 $\mu$m pore size, 13 mm in diameter, Cytiva, Marlborough, MA, USA) and bulk-electroporated with the synthetic fluorescent $Ca^{2+}$ indicator Oregon-Green 488 BAPTA-1 (OGB-1; hexapotassium salt; Thermo Fisher Scientific).

To electroporate the retina (Briggman & Euler, 2011), the anodisc was then placed between two 4 mm horizontal platinum disk electrodes (CUY700P4E/L, Nepagene/Xceltis, Mannheim, Germany). The lower electrode was covered with 15 $\mu$l of ACSF, while a 10 $\mu$l drop of 5 mM OGB-1 dissolved in ACSF covered the upper electrode and was lowered onto the tissue. Then, nine electrical pulses ($\sim$ 9.2 V, 100 ms pulse width, at 1 Hz) from a pulse generator/wide-band amplifier combination (TGP110 and WA301, Thurlby handar/Farnell, Huntingdon, UK) were applied to introduce the $Ca^{2+}$ indicator into the retinal cells. Afterwards, the retina was placed in the recording chamber, with the dorsal edge of the retina pointing away from the experimenter.

The retina was left there for $\sim$ 30 min to recover, as well as adapted to the light stimulation by displaying a binary dense noise stimulus (20 × 15 matrix, 40$\mu$m$^2$ pixels, balanced random sequence) at 5 Hz before the recordings started.

## Two-photon $Ca^{2+}$ imaging

We used a MOM-type two-photon (2P) microscope (designed by W. Denk, MPI Heidelberg; purchased from Sutter Instruments/Science Products, Novato, CA, USA; Euler et al., 2009, 2019). Briefly, the system was equipped with a mode-locked Ti:Sapphire laser (MaiTai-HP DeepSee, Newport Spectra-Physics GmbH, Darmstadt, Germany) tuned to 927 nm, two fluorescence detection channels for OGB-1 (HQ 510/84, AHF/Chroma, Tübingen, Germany) and SR101 (HQ 630/60, AHF), and a water immersion objective (CF175 LWD×16/0.8W, DIC N2, Nikon, Düsseldorf, Germany). For image acquisition, custom-made software (ScanM by M. Müller and T. Euler) running under Igor Pro 6.3 for Windows (Wavemetrics, Lake Oswego, OR, USA) was used, taking time-lapsed 64 × 64 pixel image scans ($\sim$(100 $\mu$m)$^2$) at 7.8125 Hz. A recording field comprises 100-120 cells, of which approximately 60% are responsive above threshold. Optic nerve and scan field locations were recorded to reconstruct retinal positions. For higher resolution, 512 × 512 pixel images were acquired to support semi-automatic region-of-interest (ROI) detection.

## Light stimulation

For light stimulation, we used a digital light processing (DLP) projector equipped with a lightguide port (lightcrafter DPM-E4500UVBGMKII, EKB Technologies Ltd, Bat Yam, Israel) and external UV and green light-emitting diodes (LEDs, green: 578 BP 10, F37-576; UV: 387 BP 11, F39-387; both AHF/Chroma). Visual stimuli are projected through the objective onto the retina, focusing the stimulus at the level of the photoreceptor layer (Franke et al., 2019). Both LEDs were synchronised with the microscope's scan retrace and a band-pass filter was used to further optimise the spectral separation of mouse M- and S-opsins (390/576 dual band-pass, F59-003, AHF/Chroma). Stimulator intensity (as photo-isomerisation rate, $10^3$ P/s per cone) was calibrated to range from $\sim$ 0.5 (black image) to $\sim$ 20 for M- and S-opsins, respectively. Additionally, a steady illumination of $\approx 10^4$ P/s per cone was present during the recordings due to the 2P excitation of photopigments (Euler et al., 2009, 2019).

**Table 1. Primary antibodies used**

| Target protein | Host species | Clonality | Isotype | Dilution factor | Catalogue number | Manufacturer |
|---|---|---|---|---|---|---|
| Calbindin | Guinea pig | Polyclonal | IgG | 1:500 | 214 005 | Synaptic Systems, Göttingen, Germany |
| Neurofilament H (SMI-32) | Mouse | Monoclonal | IgG1 | 1:100 | 801 701 | BioLegend, San Diego, CA, USA |

**Table 2. Secondary antibodies used**

| Host species | Target species | Isotype | Conjugated dye | Dilution factor | Catalogue number | Manufacturer |
|---|---|---|---|---|---|---|
| Goat | Guinea pig | IgG | STAR488 | 1:100 | ST488-1006-500UG | Abberior, Göttingen, Germany |
| Goat | Mouse | IgG | Alexa Fluor 568 | 1:100 | A11031 | Thermo Fisher Scientific, Waltham, MA, USA |

In total, three different light stimuli were presented: (a) a full-field chirp stimulus (700 µm in diameter; Baden et al., 2016), (b) a bright moving bar (0.3 × 1 mm) at 1 mm/s in eight directions, and (c) a random binary noise ('shifted' dense noise, SN) with a checker board grid of 20 × 15 checks and a checker size of 40 by 40 µm at 5 Hz for 5 min, which is randomly shifted on a 10 µm grid to map RFs (adapted from Pamplona et al., 2022). All stimuli were centred on the recording field. Before each stimulus, the baseline was recorded for at least 20 s after the laser scanning started to avoid immediate laser-induced effects on the retinal activity (Euler et al., 2009, 2019; Szatko et al., 2020).

### Immunohistochemistry

We used age-matched wild-type and *rd10* animals for the immunohistochemical imaging. Preparation of retinae was done in the same way as for functional imaging. Wholemount retinae were fixed with 4% paraformaldehyde (PFA) in 0.1 M phosphate buffered saline (PBS) for 20 min at 4°C. After washing the retina with 0.1 M PBS (6 × 20 min at 4°C) we blocked with 10% normal goat serum (NGS) and 0.3% Triton X-100 in 0.1 M PBS overnight at 4°C. Afterwards the primary antibodies (Table 1) in a solution with 0.3% Triton X-100 and 5% NGS in 0.1 M PBS were applied and incubated for 4–9 days at 4°C. After washing again with 0.1 M PBS (6 × 20 min at 4°C) the samples were incubated with secondary antibodies (Table 2) in solution with 0.1 M PBS overnight at 4°C. After another washing session

(6 × 20 min at 4°C) the retinae were embedded in mounting medium (Vectashield, Vector Laboratories Inc., Newark, CA, USA) on a glass slide, the dorsal side facing upwards. The samples were covered with glass coverslips (24 × 50 mm, R. Langenbrinck GmbH, Emmendingen, Germany), sealed with transparent nail polish and left to dry overnight at 4°C. Images were taken using a TCS SP8 confocal microscope (Leica Microsystems, Wetzlar, Germany) with a ×20 (NA 0.75) oil-immersion objective and an image size of 1048 × 1048 pixels. The laser was tuned to 488 and 552 nm to excite STAR488 and Alexa Fluor 568, respectively. The microscope function was controlled with LAS X software (Leica). Images were processed using the Fiji (Schindelin et al., 2012) software based on ImageJ (National Institutes of Health, Bethesda, MD, USA, (Schneider et al., 2012) and custom-made Python scripts.

### Data analysis

Data analysis was organised and performed in a custom-written schema using DataJoint for Python (http://datajoint.github.io; Yatsenko et al., 2015). Image extraction and semi-automatic ROIs detection were performed using Igor Pro 8 and custom-written Python scripts.

**Preprocessing.** After the $Ca^{2+}$ traces were extracted from individual ROIs, the raw traces were detrended by subtracting a smoothed version of the trace to remove slow drifts in the $Ca^{2+}$ signal (Szatko et al., 2020). For

smoothing, we applied a Savitzky–Golay filter (Press & Teukolsky, 1990) of third polynomial order and a window length of 60 s using the Python SciPy implementation scipy.signal.savgol (Virtanen et al., 2020).

$$r_{\text{detrend}} = r_{\text{raw}} - r_{\text{smooth}}$$

Next, the baseline activity (mean of the first 8 samples) was subtracted; the mean activity $r(t)$ was computed, and the traces were normalised such that:

$$\max_t (|r(t)|) = 1$$

**Quality filtering.** To detect reliably responding cells, two consecutive quality filtering steps were applied. To this end, the response quality index (QI) was computed for moving bar ($QI_{\text{MB}}$) and full-field chirp ($QI_{\text{Chirp}}$) responses:

$$QI = \frac{\text{Var}[\langle C \rangle_r]_t}{\langle \text{Var}[C]_t \rangle_r}$$

where $C$ is the $T$ by $R$ response matrix (time samples by stimulus repetitions) and $\langle \rangle_x$ and $\text{Var}[]_x$ denote the mean and variance across the indicated dimension $x$, respectively. Only cells with $QI_{\text{MB}} > 0.6$ or $QI_{\text{Chirp}} > 0.35$ were included in the following analyses.

**Orientation selectivity and direction selectivity.** To detect direction-selective (DS) cells, we first performed a singular values decomposition (SVD) on the normalised mean responses and projected the tuning curve on a complex exponential ($K$). The direction selectivity index (DSI) was then defined as the vector length of this vector sum:

$$DSI = |K|$$

Orientation selectivity (OS) was assessed in a similar way. For details on the calculation of DSI and OSI, see Baden et al. (2016).

**Classification of functional RGC types.** For the functional classification of RGC types, we used the method published in Qiu et al. (2023) and Gonschorek et al. (2025). This random forest-type classifier was trained, validated and tested on the RGC dataset from Baden et al. (2016). In brief, the classifier uses features of the RGC responses to the chirp and MB stimuli, the *P*-value of the permutation test (to probe direction selectivity), as well as the soma size. The confidence score (CS), which indicates the classification certainty, was used as a quality measure. We only included cells with a $CS \geq 0.25$ into the RGC type-resolved analyses.

**Receptive field estimation.** We mapped RFs using the Python toolbox RFEst (Huang et al., 2021) using the responses to the binary shifted dense noise stimulus (see 'Light stimulation'). We computed the temporal gradients of the Ca$^{2+}$ signals from the detrended traces and clipped negative values:

$$\dot{c} = \max(0, \dot{r}_{\text{detrend}})$$

The stimulus $S(t)$ and the clipped temporal gradients $c$ were up-sampled to 10 times the stimulus frequency of 5 Hz.

Spatio-temporal RFs were computed from spline-based linear Gaussian models that were optimised with gradient descent (Adam optimiser with a learning rate of 0.1) to minimise the following loss:

$$\mathcal{L} = \frac{1}{T} \int_{t=0}^{T} \left( \dot{c}(t) - y_0 X(t) Sb \right)^2 + \beta \times ||b||_1$$

where $S$ is a cubic regression spline basis, $y_0$ is the inferred intercept, $b$ are the inferred RF weights and $\beta$ is the weight for the L$_1$-penalty on $b$ to enforce sparsity in the RF.

The RF was defined as $\mathbf{F}(x, y, \tau) = Sb$, where $x$ and $y$ are the spatial dimensions and $\tau$ is the lag ranging from approx. 1.35 to $-0.20$ s. $S$ was defined by the number of knots in space and time ($k_x, k_y, k_\tau$), corresponding to the dimensions $d$ of the spatio-temporal RF ($d_x, d_y, d_\tau$) = (32, 20, 15). We optimised hyperparameters on a subset of 100 randomly drawn cells and finally set to ($k_x, k_y, k_\tau$) = (10, 12, 9) and $\beta = 0.01$.

Models were trained for at least 100 steps and a maximum of 2000 steps. If the loss did not improve for five steps, training was stopped and the parameters resulting in the lowest loss were used as the final model.

We smoothed RFs by applying a Gaussian filter of size $5 \times 5$ pixels and a standard deviation of 1 pixel to each frame. We used singular value decomposition (SVD) to decompose the RFs into a temporal $F_t(\tau)$ and spatial $\mathbf{F}_s(x, y)$ and scaled them such that $\max(|F_t|) = 1$ and $\max(|\mathbf{F}_s|) = \max(|\mathbf{F}|)$. For each spatial RF $\mathbf{F}_s$, we fit a 2D Gaussian $\mathbf{F}_{\text{Gauss}}$ using the Python package astropy (Price-Whelan et al., 2018). The area covered by two standard deviations of this Gaussian fit was used as the RF size.

Peaks in the temporal RFs were computed using scipy.signal.find_peaks setting the minimum peak height to 0.65 standard deviations of the temporal RFs. The main peak lag was defined as the lag of the peak with the smallest lag. To estimate the RF quality, we first computed a quality metric for the SVD:

$$QI_{\text{SVD}} = 1 - \frac{\text{Var}\left[ \mathbf{F}(x, y, \tau) - F_t(\tau) \mathbf{F}_s(x, y) \right]}{\text{Var}\left[ \mathbf{F}(x, y, \tau) \right]}.$$

Second, we computed a quality metric for the Gaussian fit to the spatial RF:

$$QI_{sRF} = 1 - \frac{Var\,[\mathbf{F}_s] - Var[\,\mathbf{F}_{Gauss}]}{Var\,[\mathbf{F}_s]}.$$

Only RFs with $QI_{SVD} > 0.5$, $QI_{sRF} > 0.5$ and main peak lags between 0 s and 0.3 s were used for the analysis.

**Statistical analysis.** To quantify the differences either between cell counts (Fig. 4*B*–*D*), quality measures (Fig. 5*A* and *B*) or RF properties (Fig. 8*B* and *C*), we performed the non-parametric Wilcoxon signed-rank test (Mann–Whitney *U* test) using the Python package scipy.stats (v1.11.1) and employed the multi-comparison Benjamini–Hochberg correction from the Python package statsmodels.stats.multitest (v0.14.0).

Due to large size of the dataset used for Fig. 5, which included all imaged RGCs, we additionally calculated the effect size using the rank-biserial correlation ($r_b$). This measure ranges from $-1$ to 1, with values closer to 0 indicating a weak effect, and was calculated for the Mann–Whitney *U* test as follows:

$$r_b = 1 - \frac{2U}{n_1 \times n_2}$$

where *U* is the Mann–Whitney *U* statistic, $n_1$ is the sample size of the first group, and $n_2$ is the sample size of the second group.

To test similarities between $Ca^{2+}$ traces, we used Pearson's correlation (Figs 7 and 10A).

To quantify the differences in RGC type cell numbers in the *rd10* mouse line compared to wild-type, we used a binomial test (Fig. 9*B*). We computed the total number of cells as well as the fractions per RGC type per mouse line and age. For the binomial test, we set the test parameters $k_i$ as the number of cells for each type *i* in *rd10*, and the expected proportion of cells in the corresponding RGC type, $p_i$, as the proportion of cells per type *i* in wild-type. This calculation was done for P30, P45 and P90 separately. Next, we performed a two-tailed binomial test using the Python package scipy.stats (v1.9.3). The *P*-values were corrected for false discovery rate (FDR, Benjamini–Hochberg); they were significant at the significance level of $\alpha < 0.01$.

To test similarities between distributions, we calculated Jensen–Shannon divergence (JSD; Fig. 6).

## Results

To investigate how progressive photoreceptor loss affects the functional retinal output, we employed 2P $Ca^{2+}$ population imaging in the ganglion cell layer (GCL) of *rd10* and, as a control, wild-type (C57BL/6J) *ex vivo* mouse retina. To this end, the tissue was electroporated with the fluorescent $Ca^{2+}$ indicator Oregon Green BAPTA-1 (OGB-1; Fig. 2*A*; see Methods). Here, we recorded the responses of RGCs to different light stimuli at four post-natal ages (Figs 2*B* and Fig. 3; see also Fig. 1): P30, when the retina is fully developed, rods are already significantly impacted but cones are mostly considered intact; P45, when rods are mostly lost; P90, when cones are severely damaged; and P180, when all outer-retinal photoreceptors are effectively gone (Barhoum et al., 2008; Chang et al., 2002, 2007; Gargini et al., 2007).

To evaluate if certain RGC types are more affected by photoreceptor degeneration than others, i.e. systematically or type-selectively, we presented a set of 'fingerprinting' stimuli, including full-field chirps and moving bars (Fig. 3). This allowed us to map the $Ca^{2+}$ responses to 32 groups of RGCs (Baden et al., 2016) using a previously published RGC type classifier (Gonschorek et al., 2025; Qiu et al., 2023; see Methods) trained on the dataset from Baden et al. (2016). For simplicity, in the following, we refer to these groups as 'functional RGC types'. While the classifier also identifies displaced amacrine cells (ACs), we focused our analyses on the RGCs.

Because the 2P excitation laser excites the photo-receptors (equivalent to an additional stimulus back-ground; see Euler et al., 2009, 2019), the overall stimulus levels were in the low photopic range (see also Methods). Because of this light level, it was also difficult to investigate the contribution of rod *vs*. cone photoreceptors, in particular in *rd10* at later stages of degeneration, when mostly cones are present. An earlier study (Szatko et al., 2020) has demonstrated rod-mediated responses with 2P-imaging in wild-type retinae at similar light conditions. Because in *rd10* rods are already strongly affected at P30, we here refrained from attempting to separate rod and cone responses (and used achromatic stimuli).

It has been reported that photoreceptor degeneration roughly follows a central to peripheral gradient in the *rd10* retina (Chang et al., 2007; Gargini et al., 2007), and hence we distributed our recording fields across the retina (see Fig. 2*B*) to test for such regional differences. However, in practice, the field selection was primarily guided by tissue quality and responsiveness, and therefore we refrained from analysing regional differences in RGC responses.

## RGC density remains constant throughout degeneration

To make sure that any difference observed between *rd10* and wild-type RGC responses is not confounded by a loss of RGCs during degeneration, we first determined the GCL soma density across postnatal ages. Previous studies have shown that RGCs, in comparison to cells in the

outer retina, undergo little morphological change during degeneration in *rd10* (Mazzoni et al., 2008), and hence we did not expect any dramatic differences in GCL soma density between the two mouse strains.

We used the semi-automatically drawn ROIs of each 2P recording field (Fig. 3*A* and *B*; right image column) to count the number of recorded GCL cells per post-natal age and mouse line (Fig. 4*A*). As recording fields were of constant size ($\sim$100 $\mu$m)$^2$, we used the number of cells per field as density measure (Fig. 4*B*). We found that GCL cell density slightly dropped after P30 in wild-type mice and then settled (P30 *vs.* P45, P90 and P180: $P = 0.011$, 0.0008 and 0.0048, respectively). This could be due to variability between fields but may also reflect the aftermath of developmental cell loss in the GCL (Farah & Easter Jr, 2005). Between *rd10* and wild-type, we did not find significant differences in cell density except for P30 ($P = 0.002$). Here, the intercellular space (i.e. blood vessels) was decreased in wild-type (total white area in wild-type: $N = 67, 429$ pixels; *rd10*: $N = 75, 682$ pixels); this, in combination with general variability in the recording fields, may explain this difference at P30. In any case, our data support that the progressive degeneration in *rd10* has no detectable effect on the overall RGC density. Additionally, we performed immunohistochemistry staining using Calbindin antibodies, labelling most of the cells in the GCL (Haverkamp & Wässle, 2000). Also here, we did not observe any obvious differences in cell density, supporting our findings from functional imaging (for representative examples, see Fig. 3*C*). Additionally, we analysed the density of $\alpha$-RGCs (Krieger et al., 2017), which we could identify by their distinct large soma size (soma size $> 136$ $\mu$m$^2$, see example cell marked with asterisk in Fig. 3*A*). Again, we

did not find any significant difference in density after P30 (Fig. 4*C*; for statistics, see legend). Support for our findings comes from additional immunolabelling experiments using SMI-32 (SMI) antibodies (see Methods), which strongly label $\alpha$-RGCs (Bleckert et al., 2013). We found that the density of SMI-32 positive cells in the GCL does not appear to differ between wild-type and *rd10* retinae (for representative examples, see Fig. 3*C*).

Taken together, our data suggest that the RGC density in *rd10* compared to wild-type does not change detectably during the disease progression between P30 and P180.

## Responsiveness of RGCs declines with progressing degeneration

We found that the overall RGC density was unaffected by the photoreceptor degeneration (at least until P180), implying that their morphological structure may remain largely intact. To test if they also remain functional, we next measured the quality of their light-evoked responses ('responsiveness') from P30 to P180 (Fig. 3*A* and *B* right). To this end, we first computed response quality indices (QI), which assess how reliable individual RGCs respond to the chirp and moving bar (MB) stimuli (see Methods and Baden et al., 2016). We defined a cell as responsive if $QI_{Chirp} \geq 0.35$ or $QI_{MB} \geq 0.6$.

At the earliest time point (P30), the general responsiveness did not differ between wild-type and *rd10* (Fig. 4*D*; for all statistics, see legend). However, at P45, the responsiveness of *rd10* RGCs significantly decreased compared to wild-type RGCs. This decrease continued until P180, where the percentage of responsive *rd10* cells dropped to almost zero. In contrast, the responsiveness in wild-type did not change significantly between these

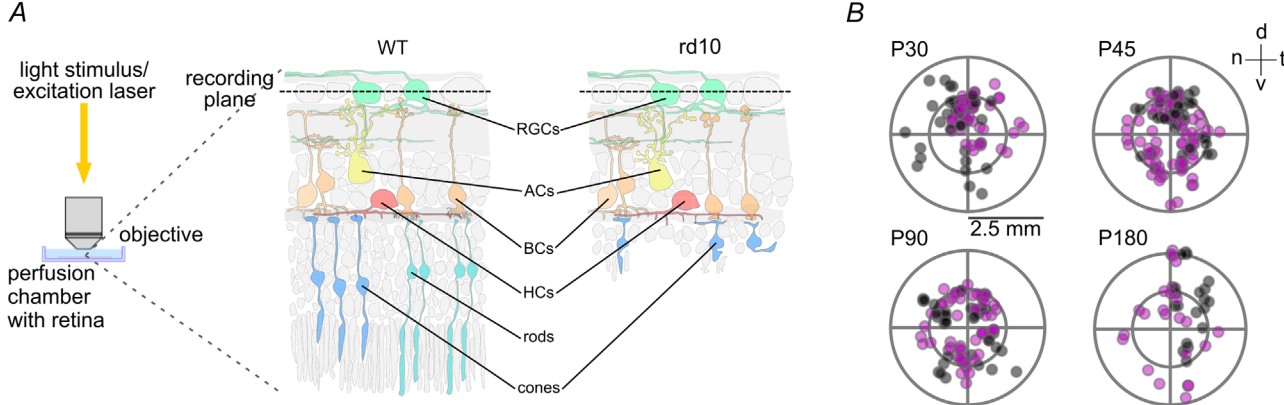

**Figure 2. Functional Ca$^{2+}$ imaging in *ex vivo* wild-type and *rd10* mouse retina**
*A*, recording configuration for two-photon (2P) imaging, with whole-mount retina in perfusion chamber and retinal ganglion cells (RGCs) facing upwards towards the objective lens (left). Illustrations of cross-sections of wild-type (WT, centre) and *rd10* (advanced degeneration, right) retina electroporated with Oregon Green BAPTA-1 (OGB-1, green). *B*, location of recording fields on the retina (circles) for the selected postnatal ages during degeneration in wild-type (black, P30, P45, P90, P180 $N = 6$, 11, 8 and 6 fields) and *rd10* (magenta, P30, P45, P90 and P180 $N = 9$, 14, 9 and 8 fields) retinae. Retinal orientation: dorsal (d), ventral (v), nasal (n), temporal (t).

ages (except for P180, Fig. 4*D*; for statistics, see legend), suggesting that the changes in responsiveness observed in *rd10* are mainly caused by degeneration.

We also computed separately the fraction of cells that responded to each of the stimuli, using the same QI thresholds as before. We reasoned that the chirp and MB stimuli challenge different aspects of retinal processing: presented full-field, the chirp mainly probes temporal properties and contrast sensitivity, whereas the MB dominantly probes spatio-temporal processing (i.e. motion sensitivity, direction selectivity). Both $QI_{Chirp}$ and $QI_{MB}$ were rather constant in wild-type across ages (with a slight increase at P180), while they steadily decreased in a similar way in *rd10* (Fig. 5*A*; for statistics, see legend). Due to the large sample sizes, we also calculated the effect

sizes for both wild-type and *rd10* (see Methods). Here, we found rather small effect sizes for wild-type, while they were medium sized in *rd10*, indicating that for wild-type, significance may be mostly due to the large sample size, while in *rd10* that was likely not the case. Accordingly, this was reflected in the fractions of cells responding to either stimulus (Fig. 5*B*). At P30, the fraction of responsive *rd10* cells was somewhat higher than in wild-type, which may be related to the higher variability in the P30 wild-type data, as mentioned above. Together, this first analysis points rather to a general, stimulus-independent decline in *rd10* RGC responsiveness.

It has been reported that the signal-to-noise ratio (SNR) of RGC responses decreases in *rd10* (Toychiev et al., 2013). We examined if we could detect this also

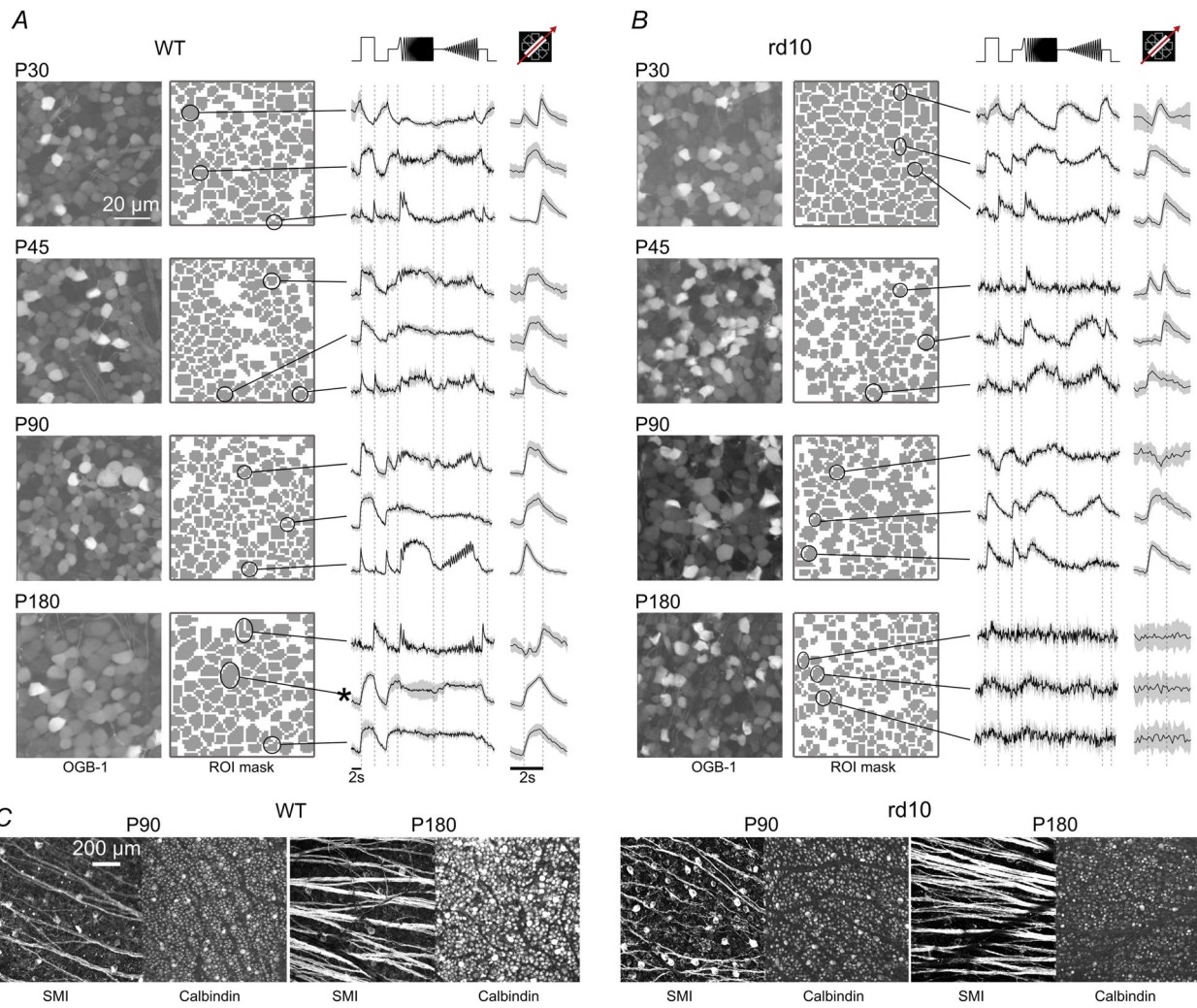

**Figure 3. Representative recordings from wild-type and *rd10* retinae**
*A*, recording fields in the ganglion cell layer (GCL) of wild-type (WT) retinae with cells loaded with fluorescent $Ca^{2+}$ indicator OGB-1 (left), corresponding region-of-interest (ROI) masks (centre), and representative responses to full-field chirps and moving bar (MB) stimuli (right) of the example cells marked in the ROI masks (continuous lines, mean responses; shaded areas, SD). Rows show examples for the four selected postnatal (P) days. Asterisk indicates α-RGC. *B*, like in *A* but for *rd10* retinae. *C*, left: representative wild-type (left) and *rd10* (right) immunohistochemical staining against SMI-32 (SMI) and Calbindin at P90 and P180.

in our data, using QI distribution as a proxy for SNR and the Jensen–Shannon divergence (JSD) to measure QI distribution differences (Fig. 6*A* and *B*). This analysis showed that QI distributions in wild-type *vs. rd10* could be quite variable between RGC types but were rather similar across ages for most functional super-groups, including 'Uncertain', 'On–Off', 'Fast On' and 'Slow On'. This suggests no substantial change in SNR in *rd10* RGCs in these groups. However, in the 'Off' group, lower JSD values indicate differences between wild-type and *rd10* for many RGC types at later degeneration stages. Combined with the observed decrease in overall responsiveness (see Fig. 4*D*) – representative of a lower QI – our data support a general decline in SNR in cells with an 'Off' component with progressing degeneration.

## The majority of RGC types remains functional until late stages of degeneration

Next, we investigated if the RGC types are systematically affected by photoreceptor degeneration or if there may be type-specific differences. To this end, we employed an RGC type classifier (Qiu et al., 2023, Gonschorek et al., 2025) to distinguish the different functional RGC types based on their response profile to the chirp and MB stimuli, their direction selectivity, as well as their soma size. This allowed us to map the cells onto the dataset published by Baden et al. (2016) (Fig. 7). As we detected

only very few responsive cells in *rd10* at P180 (Fig. 4*D*), we focused the further analyses on P30 to P90.

Throughout all investigated stages of degeneration, almost all of the previously identified 32 RGC types that the classifier was trained to detect could be found not only in wild-type but also in *rd10* (31/32/32 in wild-type *vs.* 32/32/29 in *rd10* at P30/45/90). Even at P90, we still found 29 types in *rd10*, despite the stark decline in responsive cells (Fig. 4*D*). Notably, the classifier was able to identify similar percentages of cell types across ages and mouse lines, suggesting that cell type assignment was equally reliable for both datasets (Fig. 5*D*).

To investigate if the photoreceptor degeneration affects the response profiles of distinct RGC types, we computed the correlations of their average responses to the chirp and MB between wild-type and *rd10* at each age (Fig. 7*A* and *C*). We used this correlation as a proxy for the classifier's ability to identify the known RGC types in *rd10* and, hence, as a measure for how well their responses were preserved. We found that the correlations between RGC response types in wild-type and *rd10* did not differ substantially (Figs 7*C* and 10*A*). However, distinct RGC types begin to disappear at P90 (see white vertical columns in Fig. 7*C*). For each classified RGC type, we also compared their soma size, $QI_{Chirp}$ and $QI_{MB}$ between wild-type and *rd10* and found that, overall, they matched well (Fig. 7*B*; see also Fig. 6), which implies that the progressing degeneration may not introduce strong

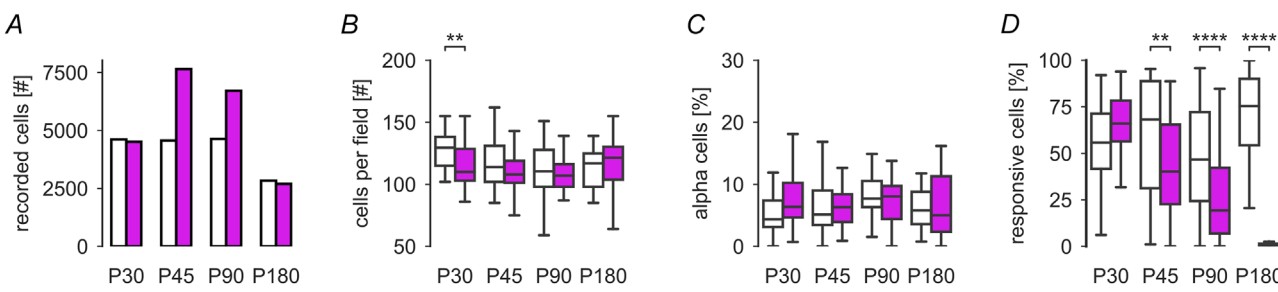

**Figure 4. RGC density and responsiveness in wild-type and *rd10* retinae from P30 to P180**
*A*, total number of recorded cells for each time point for wild-type (white bars, P30, P45, P90 and P180: $N = 4612$, 4566, 4636 and 2838 cells, respectively) and *rd10* (magenta bars, P30, P45, P90 and P180: $N = 4511$, 7649, 6719 and 2697 cells, respectively). *B*, average number of ganglion cell layer (GCL) somata per recording field for different time points (wild-type P30, P45, P90 and P180: $N = 4612$, 4566, 4636 and 2838 cells, respectively; *rd10* P30, P45, P90 and P180: $N = 4511$, 7649, 6719 and 2697 cells, respectively). Statistics for wild-type *versus rd10*, P30, P45, P90 P180: $P = 0.0022$, 0.059, 0.3415 and 0.412, respectively. *C*, percentage of $\alpha$-RGCs per recording field (wild-type P30, P45, P90 and P180: $N = 239$, 284, 388 and 163 cells, respectively; *rd10* P30, P45, P90 and P180: $N = 358$, 532, 479 and 275 cells, respectively). Statistics for wild-type *versus rd10*, P30, P45, P90 and P180: $P = 0.023$, 0.494, 0.291 and 0.741, respectively. For wild-type, the number of $\alpha$-cells differs for P90 from all other time points (*vs.* P30, P45 and P180: $P = 0.020$, 0.0088 and $8.091 \times 10^{-63}$, respectively). *D*, percentage of responsive cells per recording field (see Methods, wild-type P30, P45, P90 and P180: $N = 2532$, 2691, 2448 and 2029 cells, respectively; *rd10* P30, P45, P90 and P180: $N = 2790$, 3206, 1793 and 30 cells, respectively). Statistics for wild-type *versus rd10*, P30, P45, P90 and P180: $P = 0.106$, 0.0044, $7.545 \times 10^{-5}$ and $1.701 \times 10^{-9}$, respectively. For wild-type, the number of responsive cells is stable except for an increase in responsiveness between P30 and P180 ($P = 0.027$). Statistics for *rd10* RGCs, P30 *versus* P45, P90 and P180: $P = 9.98 \times 10^{-5}$, $2.4 \times 10^{-9}$ and $1.43 \times 10^{-10}$, respectively; P45 *versus* P90 and P180: $P = 8 \times 10^{-4}$ and $1.34 \times 10^{-10}$, respectively; P90 *versus* P180: $P = 6.62 \times 10^{-10}$. *B*–*D*: Mann–Whitney with Benjamini–Hochberg correction, asterisks indicate significance levels for comparisons between mouse lines: **$P < 0.001$, ****$P < 0.0001$.

functional differences between the RGC types of both mouse strains.

Finally, we compared the correlation for each RGC type within the two mouse lines. Here, we found that it generally varies somewhat over the three ages, suggesting that there may be intrinsic variability in both mouse lines, which is potentially not related to degeneration.

Note that a classifier-based approach cannot discover potentially new degeneration-related response types (see Discussion). Still, our type-resolved analysis clearly suggests that the functional RGC diversity in *rd10* is largely comparable to that in wild-type at least until P90.

Our findings highlight the robustness of RGC diversity in *rd10* despite degeneration.

### Receptive fields tend to be smaller and have faster kinetics in *rd10*

Photoreceptor degeneration has been reported to affect RGC receptive field (RF) properties, such as RF size (Care et al., 2020; Ellis et al., 2023; Scalabrino et al., 2022). Therefore, we estimated spatio-temporal RFs in both mouse lines using the responses to a shifted dense

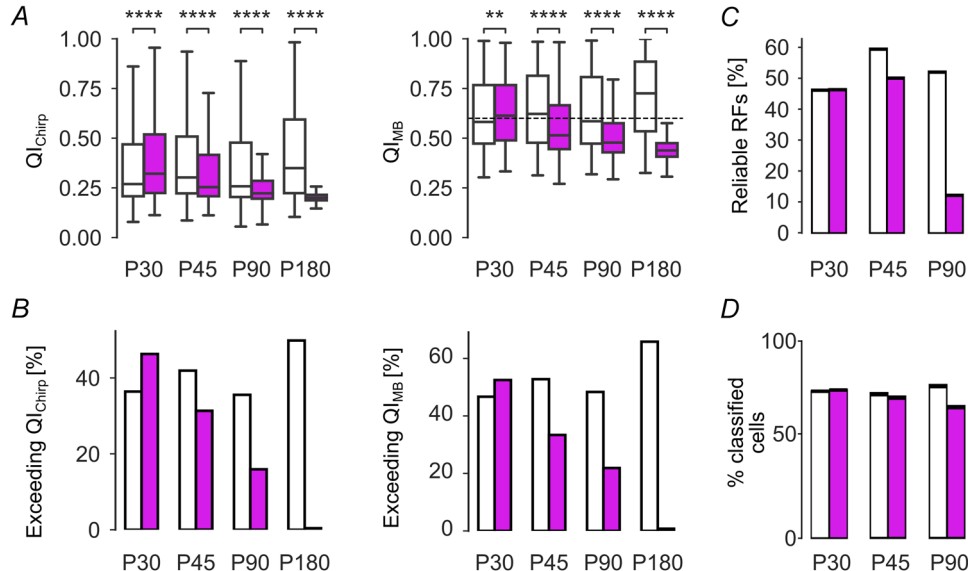

**Figure 5. Quality of the light-evoked responses**

*A*, left: distribution of quality index (QI) for chirp responses. Statistics for wild-type RGCs (white bars): P30 *versus* P45, P90 and P180: $P = 5.28 \times 10^{-17}$, $1.88 \times 10^{-1}$ and $3.66 \times 10^{-33}$, respectively; P45 *versus* P90 and P180: $P = 3.56 \times 10^{-22}$ and $3.58 \times 10^{-8}$, respectively; P90 *versus* P180: $1.45 \times 10^{-37}$. Effect sizes for wild-type, P30 *versus* P45, P90 and P180: 0.115, −0.024 and 0.141, respectively; P45 *versus* P90 and P180: −0.138 and 0.036, respectively; P90 *versus* P180: 0.160. Statistics for *rd10* RGCs (magenta bars), P30 *versus* P45 and P90: $P = 1.64 \times 10^{-63}$ and $7.85 \times 10^{-295}$, respectively; P45 *versus* P90: $4.00 \times 10^{-124}$; P90 *versus* P180: $2.07 \times 10^{-169}$. Effect sizes for *rd10*, P30 versus P45, P90 and P180: −0.178, −0.379 and −0.689, respectively; P45 *versus* P90 and P180: −0.209 and −0.551, respectively; P90 *versus* P180: −0.360. Statistics for wild-type *versus rd10*, P30, P45 and P90: $4.86 \times 10^{-28}$, $3.86 \times 10^{-43}$ and $2.792 \times 10^{-93}$, respectively. Statistics for P30, P45 *versus* P180 are not reported, because the large differences between response quality resulted in *P*-values approaching zero. Right: same as on the left but for moving bar (MB) response QI. Statistics for wild-type, P30 *versus* P45, P90 and P180: $N = 1.22 \times 10^{-7}$, $2.08 \times 10^{-2}$ and $1.64 \times 10^{-83}$, respectively; P45 *versus* P90 and P180: $7.36 \times 10^{-3}$ and $5.45 \times 10^{-48}$, respectively; P90 *versus* P180: $2.7 \times 10^{-63}$. Effect sizes for wild-type, P30 *versus* P45, P90 and P180: 0.079, 0.023 and 0.249, respectively; P45 *versus* P90 and P180: −0.052 and 0.169, respectively; P90 *versus* P180: 0.218. Statistics for *rd10*, P30 *versus* P45 and P90: $P = 1.12 \times 10^{-112}$ and $5.72 \times 10^{-283}$, respectively; P45 *versus* P90: $1.52 \times 10^{64}$; P90 *versus* P180: $1.21 \times 10^{-155}$. Effect sizes for *rd10*, P30 *versus* P45, P90 and P180: −0.238, −0.376 and −0.683, respectively; P45 *versus* P90 and P180: −0.147 and −0.482, respectively; P90 *versus* P180: −0.340. Statistics for wild-type *versus rd10*, P30, P45 and P90: $P = 0.00115$, $1.224 \times 10^{-117}$ and $2.544 \times 10^{-231}$, respectively. Statistics for P30 and P45 *versus* P180 are not reported, because the large differences between response quality resulted in *P*-values approaching zero. *B*, left: percentage of RGCs exceeding the QI threshold for chirp responses. *rd10*: P30, 46.3%; P45, 31.3%; P90, 15.9%; P180, 0.4%; wild-type: P30, 36.4%; P45, 42.9%; P90, 35.6%; P180, 49.9%. Right: same as on the left but for moving bar (MB) responses. *rd10*: P30, 52.5%; P45, 33.4%; P90, 21.9%; P180, 0.8%; wild-type: P30, 46.7%; P45, 52.8%; P90, 48.3%; P180, 65.8%; *C*, percentage of cells that passed the receptive field (RF) quality threshold (see Methods) from all classified RGCs, i.e. cells that passed the QI thresholds for chirp and MB responses, and were assigned an RGC type, for wild-type (white) and *rd10* (magenta) RGCs. *D*, percentage of classified cells that pass our confidence score (CS) thresholds (CS ≥ 0.25 *vs.* no CS; see Methods) for wild-type and *rd10* RGCs.

noise stimulus (Pamplona et al., 2022), which allowed us to probe the cells' spatial and temporal properties in more detail. To this end, we used generalised linear models (GLMs; see Methods). As a quality measure, we only used RFs that passed the three quality thresholds: $QI_{SVD} > 0.5$, $QI_{sRF} > 0.5$ and $QI_{tRF} > 0.85$ (for details, see Methods).

We found good-quality RFs for both wild-type and *rd10* RGCs at all three time points, with the number of RFs declining in *rd10* at P90 (Fig. 5*C*). This decline may be at least partially cell type-specific, as for some functional types (e.g. 'On–Off local', $G_{11}$) we did not find RFs at P90 in *rd10* (Fig. 8*A*).

First, we investigated the size of spatial RFs (sRFs, Fig. 8*B*) and found that *rd10* RF centres tended to be smaller. Specifically, at P30, *rd10* sRFs were 5.3% smaller than wild-type RFs (*rd10*: mean, 120.2 μm *vs.* wild-type: 126.9 μm, *P* = 0.0014). Similarly, at P45, *rd10* sRFs were 4.3% smaller (*rd10*: 117.2 μm *vs.* wild-type: 122.5 μm, *P* = 0.013), while at P90, this difference was minimal (0.2%; *rd10*: 128.9 μm *vs.* wild-type: 129.2 μm, *P* = 0.829; Fig. 8*B*). Interestingly, sRFs varied across and within cell types: For example, for $G_{22}$ ('On-sustained') cells, sRF sizes were consistently smaller in *rd10 vs.* wild-type at all ages (from 13.6%, at P30 to 4.0% at P90), but the differences were not statistically significant.

Next, we quantified the temporal RF (tRF, Fig. 8*C*) properties by computing the time lag to the main $Ca^{2+}$ event ($\Delta t$ of the kernel peak closest to zero; see Methods), as a measure of tRF kinetics. This analysis revealed that overall, tRFs tended to be faster in *rd10* compared to wild-type. At P45, *rd10* tRFs were 15.4% faster than in wild-type (*rd10*: 0.11 s *vs.* wild-type: 0.13 s, *P* = $4 \times 10^{-8}$; Fig. 8*C*, right, centre). Also, at P90, tRFs tended to be faster in *rd10* (10.3%; *rd10*: mean, 0.113 s *vs.* wild-type: 0.126

s, *P* = 0.504). However, this trend was not uniform, with substantial heterogeneity among cell types. For instance, in $G_{17}$ ('On local transient') cells, the most significant differences in tRF kinetics were observed at P45, where *rd10* cells exhibited a 15.4% faster kinetics than wild-type cells (*rd10*: 0.13 s *vs.* wild-type: 0.11 s, p = 0.0082), whereas at P30 and P90, no significant differences were observed (P30: 0.12 s *vs.* 0.12 s, *P* = 0.779; P90: 0.113 s *vs.* 0.126 s, *P* = 0.5043; Fig. 8*C*, left).

Taken together, our RF analyses showed cell type-specific differences in sRFs and tRFs between *rd10* and wild-type mice. Given the high degree of cell-to-cell variability, interpreting the population-level trends is difficult. This variability might reflect intrinsic differences between functional cell types and/or differences in their susceptibility to progressing degeneration (e.g. changes in light-driven synaptic input and/or inner retinal circuits). While these differences were quite heterogeneous and time point-dependent, we overall found that *rd10* RFs were smaller, especially at P30 and P45, and at least at P45, *rd10* exhibited faster kinetics. These findings suggest that degeneration in *rd10* may impact spatial and temporal RF properties differently across cell types, with some functional types showing more pronounced differences, particularly in earlier stages of degeneration.

## Progressive, type-specific degeneration of RGCs in the *rd10* retina

So far, our analyses showed that the majority of RGC types can be found in both mouse lines at least until P90, suggesting that the overall functional diversity remains largely unaffected in *rd10*. On the other hand, we found changes in RF properties early in the degeneration.

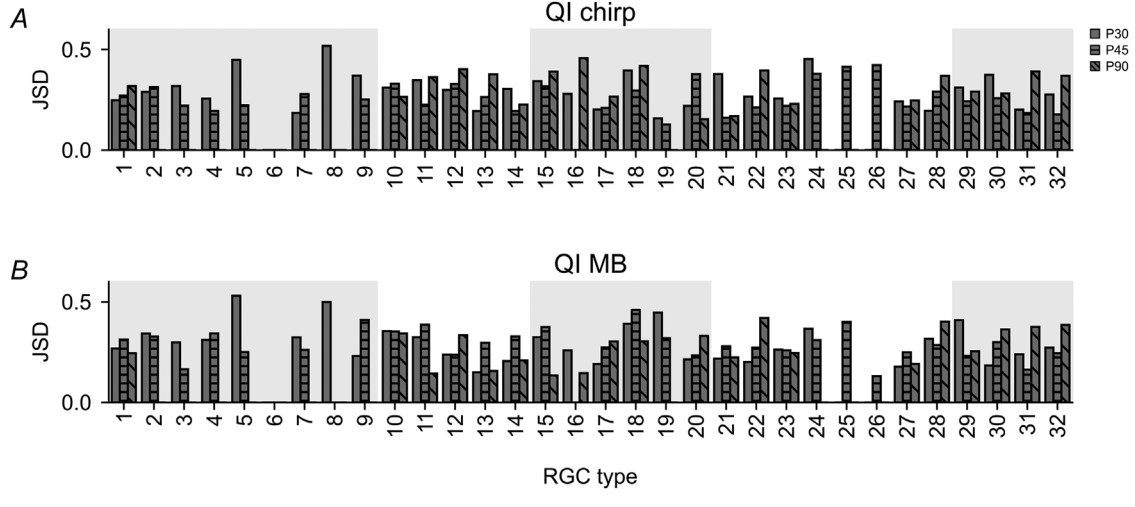

**Figure 6. Similarity between wild-type and *rd10* quality index (QI) distributions**
*A*, Jensen–Shannon divergence (JSD) between wild-type and *rd10* QI distributions of chirp responses for each cell type. 'Empty' columns had too few cells to compute JSD (P30: solid fill; P45: horizontal hatching; P90: diagonal hatching). *B*, same as in *A*, but for QI distributions of moving bar (MB) responses.

Therefore, we next asked if the relative composition of RGC types is different in *rd10* compared to wild-type (Fig. 9).

We compared the percentage of responsive cells of all recorded cells per RGC type across mouse lines and ages (Fig. 9*A*). This confirmed that nearly all RGC types were present in wild-type and *rd10*, and match the response patterns reported by Baden et al. (2016) quite well (Fig. 10*A*). However, we were surprised to see that some RGC types were under-represented (e.g. $G_{1,4}$), while others were overrepresented (e.g. $G_{17,32}$) in *rd10 vs.* wild-type. To better visualise these differences, we calculated an index that captures the 'relative abundance' of RGC types ($\log_2(rd10/WT)$; Fig. 9*B*). This measure has previously been introduced to quantify genetic RGC

types regarding their resilience to injury (Tran et al., 2019) and the contribution of RGC types to dLGN responses (Román Rosón et al., 2019). In our study, positive values indicate higher, negative values lower abundance in *rd10*, suggestive of more resilient or more vulnerable functional types, respectively.

To our surprise, the distribution in *rd10* deviated significantly from that in wild-type as early as P30 (Fig. 9*B*, top). Compared to wild-type, we found fewer 'Off' and 'On–Off' cells in *rd10*, whereas the relative fraction of 'On' RGCs (both 'Fast On' and 'Slow On') and distinct 'Uncertain' RGCs (i.e. $G_{29}$) was larger in *rd10*. At P45, most of the 'Slow On' cell types were less abundant in *rd10* (Fig. 9*B*, centre), with the exception of $G_{21}$ ('On low frequency'), which was more frequent. Like at P30,

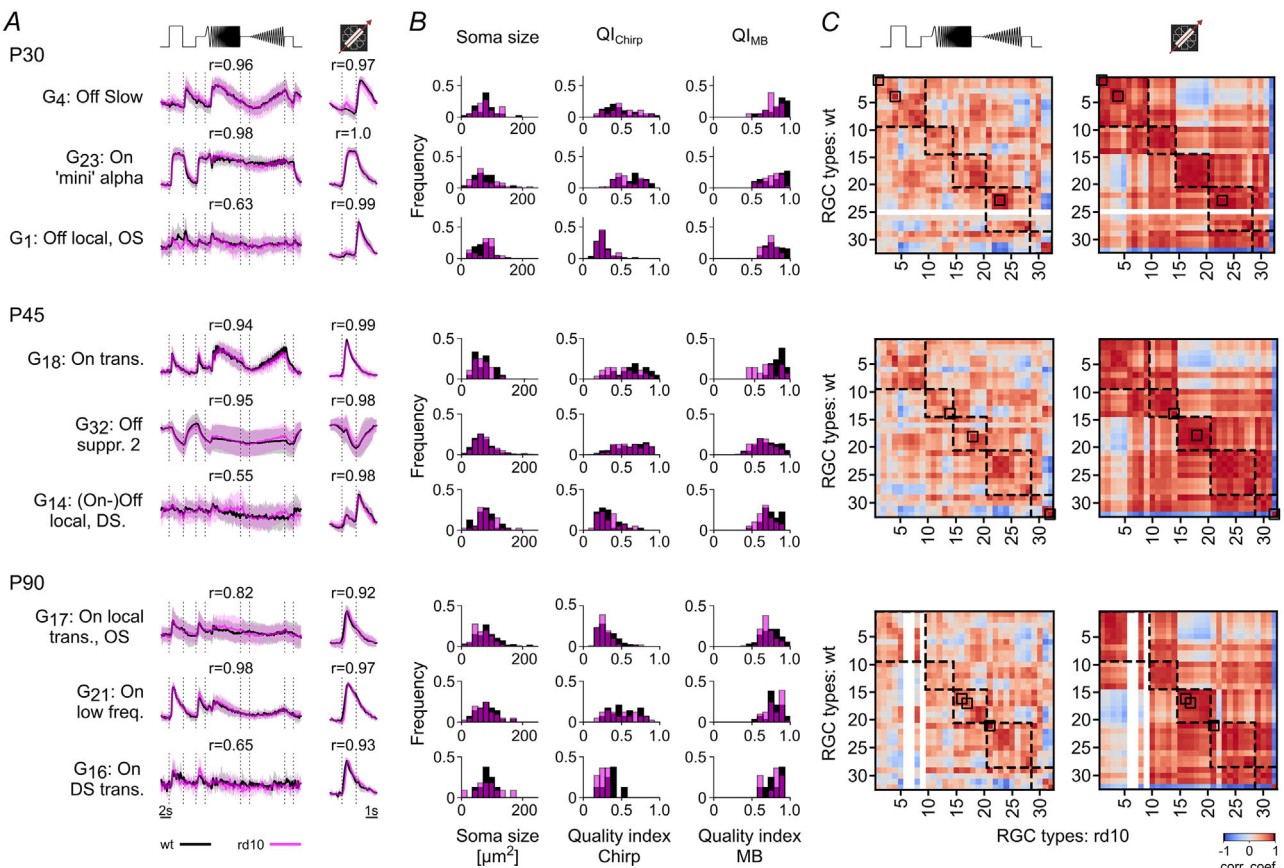

**Figure 7. Classifying and comparing RGC responses of P30, P45 and P90, between wild-type and *rd10* retinae**

*A*, representative RGC type responses to chirp (left) and moving bar (MB, right) stimuli (black, wild-type; magenta, *rd10*) for three ages (top, P30; middle, P45; bottom, P90). *r*-values above traces indicate Pearson correlation coefficients between the average RGC type responses in wild-type and *rd10* retinae. *N*-values for wild-type and *rd10* respectively: $G_4$, 39/18; $G_{23}$, 30/52; $G_1$, 59/28; $G_{18}$, 21/28; $G_{32}$, 237/440; $G_{14}$, 40/39; $G_{17}$, 128/103; $G_{21}$, 29/41, and $G_{16}$, 8/11 cells. *B*, distributions of three parameters (soma size, $QI_{Chirp}$, $QI_{MB}$) of the corresponding RGC types in *A* for *rd10* (magenta) and wild-type (black). *C*, correlation matrix of type average responses per RGC type between wild-type and *rd10* for chirp (left) and MB (right), with colour encoding Pearson correlation coefficient. Dashed boxes highlight the functional super-groups: 'Off', 'On–Off', 'Fast On', 'Slow On' and 'Uncertain'; see Baden et al. (2016). Black boxes within the correlation matrices indicate the corresponding RGC types from *A*. White stripes indicate missing RGC types in the corresponding mouse line.

the 'Fast On' RGCs tended to be more abundant in *rd10*; this changed at P90, when only G$_{15}$ ('On step') was more frequent among the 'Fast On' RGCs in *rd10* (Fig. 9*B*, bottom).

To illustrate cell type-specific changes in relative abundance, we highlight three representative examples: G$_1$ ('Off local, OS'), G$_{13}$ ('On–Off DS 2') and G$_{27}$ ('On slow'). Across all ages, G$_1$ cells exhibited a markedly reduced relative abundance in *rd10* compared to wild-type already at P30 and more pronounced at P45 and P90 (Fig. 9*B*, from top to bottom). This is

suggestive of a significant and consistent vulnerability of this type to photoreceptor loss. In contrast, G$_{13}$ cells showed a more transient pattern of change. At P30, its relative abundance was significantly reduced in *rd10*, while at P45 and P90, no significant differences were observed. This pattern may indicate a higher resilience and/or compensatory mechanisms in the upstream circuitry at later degeneration stages. Other than the first two examples, G$_{27}$ displayed no significant change in abundance in *rd10 versus* wild-type.

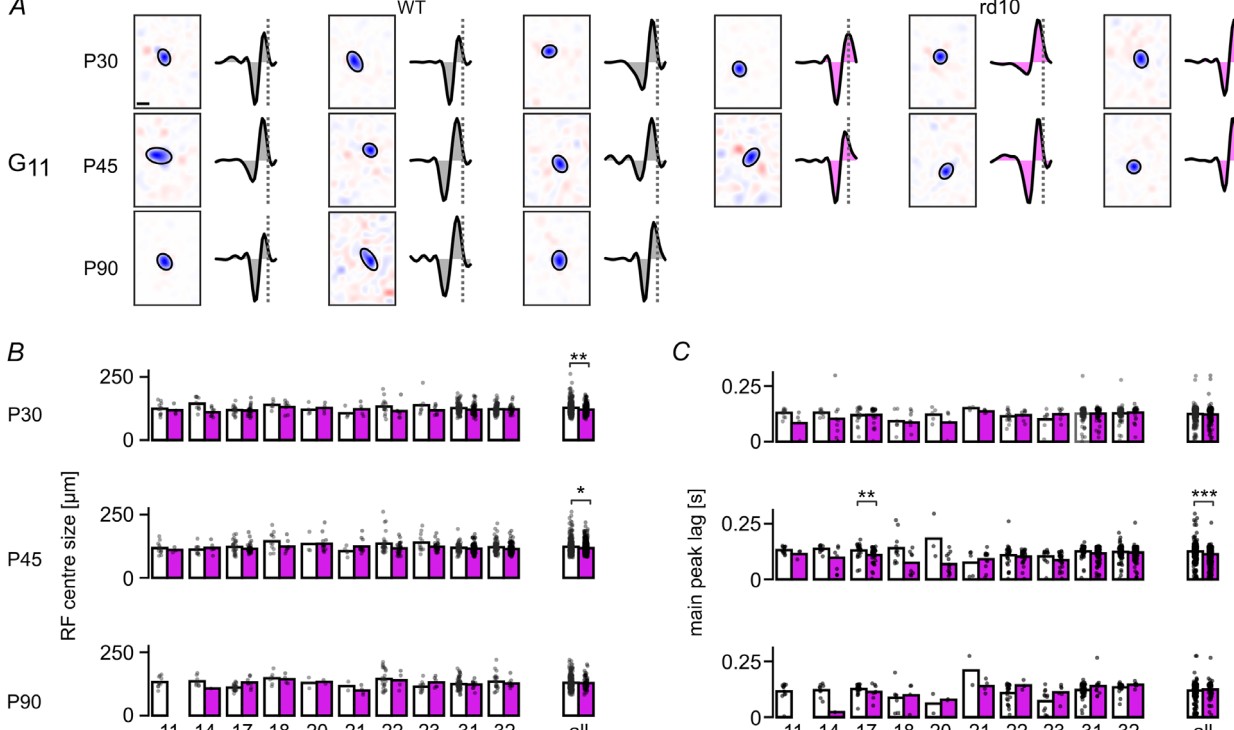

**Figure 8. Receptive field properties in *rd10 versus* wild-type retina**
*A*, representative spatial and temporal receptive fields (sRF and tRFs, respectively) of 'On–Off local' (G$_{11}$) RGCs for different ages (rows) and mouse lines (wild-type, white, left; *rd10*, magenta, right). sRF centre indicated by Gaussian fit (ellipse; scale bar: 50 μm). Dotted line indicates onset of Ca$^{2+}$ event in tRF (scale bar: 0.5 s). *B*, left: centre size of sRF for different RGC types in wild-type (white, P30, G$_{11, 14, 17, 18, 20, 21, 22, 23, 31, 32}$ and all: *N* = 12, 10, 19, 7, 5, 4, 17, 8, 111, 93 and 353 cells, respectively; P45: *N* = 13, 11, 27, 11, 3, 5, 30, 12, 52, 97 and 372 cells; P90: *N* = 11, 9, 25, 7, 2, 2, 25, 12, 70, 24 and 263 cells, respectively) and *rd10* (magenta, P30 G$_{11, 14, 17, 18, 20, 21, 22, 23, 31, 32}$ and all *N* = 4, 9, 25, 8, 4, 5, 7, 8, 63, 30 and 185 cells, respectively; P45: *N* = 3, 6, 46, 8, 14, 13, 28, 29, 139, 162 and 507 cells; P90: *N* = 0, 1, 8, 4, 2, 4, 4, 6, 12, 10 and 63 cells, respectively) as mean (bar) and for individual cells (dots). Only types with *N* ≥ 4 responsive cells per mouse line at P30 are shown. Right: centre size of all RGC types (including types not shown on the left) as mean (bar) and for all cells (dots). Statistics for wild-type *versus rd10*, G$_{11, 14, 17, 18, 20, 21, 22, 23, 31, 32}$ and all, P30: *P* = 0.8113, 0.0623, 0.8113, 0.8113, 0.8113, 0.8113, 0.1844, 0.5862, 0.1542, 0.8574 and 1.441 × 10$^{-3}$, respectively; P45: *P* = nan, 0.6605, 0.3380, 0.3380, nan, 0.3380, 0.1822, 0.3380, 0.3380, 0.1822 and 1.334 × 10$^{-2}$; 'nan'-values indicate too few cells in the respective comparisons. *C*, same as *B* but for main peak lag of the tRF (i.e. time to Ca$^{2+}$ event). Statistics for wild-type *versus rd10*, G$_{11, 14, 17, 18, 20, 21, 22, 23, 31, 32}$ and all, P30: *P* = 0.1587, 0.2354, 0.7789, 0.7789, 0.5896, 0.1587, 0.7789, 0.4127, 0.3241, 0.2022 and 6.1660 × 10$^{-1}$, respectively; P45: *P* = nan, 0.1251, 0.0082, 0.1090, nan, 0.9241, 0.5699, 0.1134, 0.1040, 0.7553 and 3.838 × 10$^{-8}$; P90: *P* = nan, nan, 0.5043, 0.6485, nan, nan, 0.4064, 0.2004, 0.2004, 0.4230, 0.2004 and 4.0967 × 10$^{-1}$, respectively; 'nan'-values indicate too few cells in the respective comparisons. *B* and *C*: Mann–Whitney. On the RGC type level Benjamini–Hochberg correction was used. Tests were only calculated for *N* ≥ 4 in both groups; asterisks indicate significance levels: **P* <0.05, ***P* <0.01 and ****P* <0.005.

These examples underscore the main trajectories of functional RGC types we observed in response to photoreceptor degeneration: (i) many types show a more or less consistent decrease in abundance (e.g. $G_1$, $G_4$, $G_{10}$); (ii) some exhibit more complex changes, with shifts in abundance occurring at different stages of degeneration (e.g. $G_{13}$, $G_{20}$, $G_{29}$); and finally (iii) other types show little or no change in their abundance (e.g. $G_{19}$, $G_{24}$, $G_{27}$). Together, this variability suggests that retinal degeneration affects different RGC types to varying extents and potentially through different mechanisms (see Discussion).

Notably, overall, the relative abundance of RGCs with a similar response type (i.e. functional super-groups) changed similarly over the course of degeneration

(Fig. 9C). The relative abundance became negative for 'Off' and somewhat less so for 'On–Off' RGCs already at P30, whereas for 'Slow On' cells, it dropped below zero for P45 and later. For 'Fast On' RGCs, the relative abundance became negative only at P90, while that for 'Uncertain' RGCs stayed close to zero for all ages. This suggests differences in resilience among functional RGC super-groups (from resilient to vulnerable): 'Uncertain' > 'Fast On' > 'Slow On' > 'On–Off' > 'Off'.

Finally, we evaluated if the photoreceptor degeneration affected orientation selectivity and direction selectivity (OS, DS; Fig. 9D). To this end, we divided RGC types into OS/Non-OS (Fig. 9D, left) and DS/Non-DS (Fig. 9D, right; see Methods). We found that the relative abundance of OS and Non-OS cells declined similarly, indicating that the

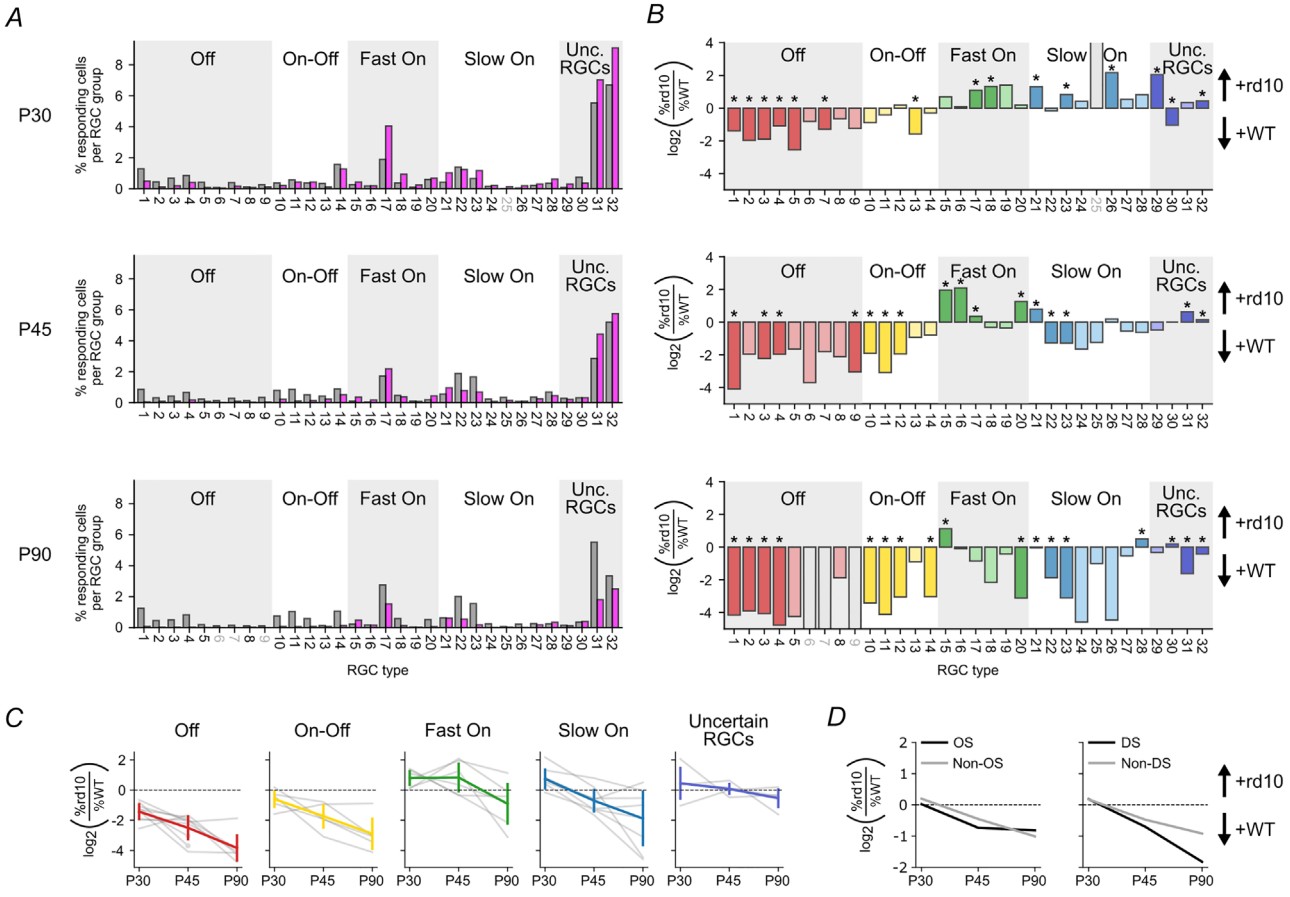

**Figure 9. Progressive, type-specific degeneration in the *rd10* retina**
*A*, distribution of responsive cells per functional RGC type (in percent) for wild-type (WT, grey; P30, P45, P90 and P180: *N* = 4612, 4566, 4636 and 2838 total cells, respectively) and *rd10* (magenta; P30, P45, P90 and P180: *N* = 4511, 7649, 6719 and 2697 total cells, respectively) across three different ages (from top to bottom: P30, P45 and P90). *B*, relative abundance of functional RGC types in *rd10 versus* wild-type, quantified as $\log_2(rd10/WT)$ for the three ages. Colours indicate functional super-groups ('Off', 'On–Off', 'Fast On', 'Slow On', 'Uncertain RGCs'). Positive values indicate higher abundance in *rd10*, negative values a higher abundance in wild-type. Significant differences in cell type ratios between *rd10* and wild-type ($P < 0.01$; binomial test) are marked with asterisks; non-significant RGC types are coloured in muted colours. Open bars indicate missing types in either mouse line. *C*, relative abundance for the five functional RGC super-groups (as in *B*) as a function of age (grey, individual RGC types; coloured, mean ± SD). *D*, relative abundance for orientation-selective (OS) and Non-OS (left) and direction-selective (DS) and Non-DS RGC types (right) as a function of age.

loss in OS responses may simply reflect the general loss in responsiveness of OS RGCs. In contrast, the relative abundance of DS cells dropped faster than that of Non-DS cells, which may be related to the 'On–Off' RGCs being more susceptible to the degeneration (Fig. 9*B* and *C*).

Taken together these results indicate that functional RGC types may undergo a progressive, type-specific degeneration in *rd10*.

## Discussion

Learning how degenerative diseases differentially affect functional cell types will provide a greater understanding of disease progression and may point toward novel therapeutic approaches. In this study, we measured light-evoked responses RGCs in the *rd10* mouse from P30 – when the retina is adult, rods are already impacted but cones are intact – to virtually complete photoreceptor loss at P180. We found that during this time window, the overall number of cells in the GCL was not affected by photoreceptor degeneration and comparable to the wild-type situation. To investigate how retinal output changes during disease progression, we used 2P $Ca^{2+}$ imaging to record RGC responses to various visual stimuli in the low photopic range. Here, we found that almost all known functional RGC types can be identified even at a late degeneration state (P90), though with slight changes in RF properties in *rd10*. Critically, however, the relative abundance of specific RGC types decreased in *rd10 vs*. wild-type in a degeneration stage-dependent

manner, starting with a significant loss of 'Off' RGC types already at P30.

## *rd10* as a model for photoreceptor degeneration in retinitis pigmentosa

Photoreceptor degeneration in the *rd10* mouse is caused by a missense mutation in the $\beta$-subunit of the phosphodiesterase gene (Chang et al., 2000). Rod degeneration begins around P16, shortly after eye opening (P14), with the peak occurring around P20. By P45, nearly all rods are lost, and secondary cone degeneration has begun. At least until P60, cones can still generate light-evoked responses and drive their circuitry at photopic light levels (Ellis et al., 2023). Around P90, the remaining cones are severely deformed, and by P180, they are almost completely lost (for an overview, see Fig. 1, and see Barhoum et al., 2008; Gargini et al., 2007; Puthussery et al., 2009).

To some extent, *rd* mouse mutants, such as *rd10*, mimic human forms of autosomal recessive RP (Chang et al., 2000, 2002), making them suitable for studying this condition. For example, many *rd* mice exhibit spontaneous oscillatory activity after rod loss (Goo et al., 2011; Haq et al., 2014; Haselier et al., 2017; Stasheff, 2008), altering the signal-to-noise ratio (SNR), and complicating therapeutic interventions (e.g. Stutzki et al., 2016). Notably, these oscillations may be related to photopsia, a phenomenon observed in human RP patients,

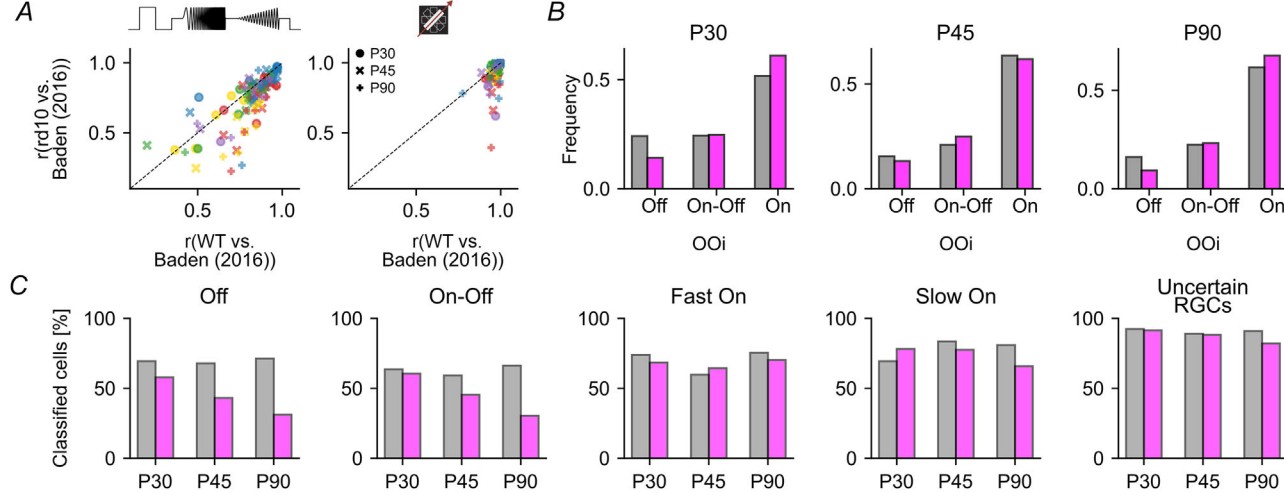

**Figure 10. RGC type classifier performance**
*A*, scatter plot of Pearson's correlation coefficients (*r*) between our datasets and the wild-type (WT) dataset published by Baden et al. (2016), with *rd10 versus* Baden et al. (2016) on the *y*- and wild-type *versus* Baden et al. (2016) on the *x*-axis, for chirp (left) and moving bar stimulus (right). Data points represent RGC type, with colour indicating the functional super-group (red: Off; yellow: On–Off; green: Fast On; blue: Slow On; purple: Uncertain RGCs). *B*, frequency of On–Off index (OOi) per age and mouse line (grey: wild-type; magenta: *rd10*) for three bins (Off: $-1 < OOi \leq -0.2$; On–Off: $-0.2 < OOi \leq 0.2$; On: $0.2 < OOi \leq 1$). *C*, percentage of cells classified with confidence score (CS) threshold $\geq 0.25$ compared to no CS threshold (per super-group, age and mouse line).

which hampers their quality of life (Bittner et al., 2009; reviewed in Stasheff, 2018).

One advantage of *rd10 vs.* models like *rd1* (Pittler & Baehr, 1991) is the slower disease progression, which is more similar to that in human but still allows experimental studies in a reasonable time frame. On the other hand, RP typically starts between teenage years and young adulthood (Hamel, 2006; Hartong et al., 2006), while in *rd10*, retinal development is not concluded when photoreceptor degeneration starts. This is most notable from the fact that *rd10* electroretinograms (ERGs) are never normal (Chang et al., 2002), already starting with decreased sensitivity at low light levels (Chang et al., 2007). Newer mouse models, such as *Cngb1*$^{neo/neo}$ (Wang et al., 2019), exhibit a slower progression, which may mimic more closely the time course of human RP.

Another difference between *rd* mice and humans is the spatial gradient of degeneration: while human retinal degeneration typically starts peripherally and progresses toward the centre, leading to the initial loss of peripheral vision (Hartong et al., 2006), degeneration in *rd10* mice begins more centrally (Chang et al., 2007; Gargini et al., 2007). In our dataset, it is difficult to evaluate this gradient, because the recording field selection was biased to responsive retinal regions (see Results), as our main goal was to study the degeneration effects on RGC light responses.

In summary, the *rd10* mouse model offers significant advantages for studying retinal degeneration, particularly due to its genetic relevance and well-documented disease progression. However, its limitations, such as differences in the spatial disease progression (i.e. in the context that mice lack a fovea) and the overlap between retinal development and degeneration onset, must be considered when interpreting results.

## Effects of photoreceptor degeneration on retinal organisation

Photoreceptor degeneration goes hand in hand with extensive remodelling in the outer retina of *rd* mice (Strettoi et al., 2003). This remodelling occurs particularly at the level of bipolar cells (BCs). Rod bipolar cells (RBCs) form new contacts with remnant cones after losing their synaptic input (Peng et al., 2000), while cone bipolar cells (CBCs) extend their branches to establish targeted connections with surviving cones after losing their original synaptic partners (Strettoi et al., 2004). However, these structural changes do not seem to include RGCs, with their synaptic organisation, central projections, and numbers being comparable to wild-type (Mazzoni et al., 2008). For instance, even as their synaptic partners, type 6 CBCs, alter their dendritic connectivity, 'On-$\alpha$' RGCs do not change their wiring or function (Care

et al., 2019). Still, it stands to reason that the inner retina eventually changes with progressing degeneration.

Indeed, recent studies on photoreceptor loss in *rd* mice have found subtle changes in RGC light responses already early in degeneration. For example, RFs shrink, and firing rates as well as information rates decrease, in particular at low light levels (Ellis et al., 2023; Scalabrino et al., 2022). In line with these results, we found *rd10* RFs to be smaller, partially with faster kinetics compared to wild-type. Similarly, reductions in RF sizes were reported in rat RP models, though, other than our findings, RF kinetics became slower in a somewhat type-specific manner (Sekirnjak et al., 2011; Yu et al., 2017). Generally, changes in RFs seem to be strongly model- and insult-dependent. For example, regarding different types of $\alpha$-RGCs, partial ablation of cones, on the one hand, resulted in 'On-$\alpha$-sustained' RGCs displaying larger RF sizes and slower kinetics (Care et al., 2019). A study of induced ocular hypertension, on the other hand, found 'Off-$\alpha$-transient' but not 'On-$\alpha$-sustained' cells showing a decrease in RF size (Ou et al., 2016).

These studies, together with our results, suggest that RGCs retain their functionality late into degeneration. At the same time, RGCs are affected by the remodelling – as we observed, for example, nuanced effects on temporal and spatial processing. Importantly, we found these differences to be cell type-specific and time point-dependent (see next section).

At later stages, when photoreceptor loss is advanced, RGCs display spontaneous oscillatory activity in various RP models, including *rd1* and *rd10* (Goo et al., 2011; Haq et al., 2014; Stasheff, 2008). This oscillatory activity has been shown to originate, at least in part, from the AII amacrine cell–CBC network (Choi et al., 2014; Trenholm et al., 2012), which increasingly lacks light-evoked synaptic input from the outer retina (Biswas et al., 2014; Margolis et al., 2008; Stasheff et al., 2011). Therefore, this spontaneous activity may also reflect inner retinal circuit alterations. In our experiments, we were not able to see the oscillatory activity at the RGC level, likely because in *rd10*, its frequency is around 10 Hz (e.g. Goo et al., 2011; Margolis et al., 2008), which we cannot resolve at our scan rate of 7.8125 Hz. The spontaneous activity was reported to interfere with RGC responses, for instance, to electrical stimulation (Ahn et al., 2022; Haselier et al., 2017). While we used light stimulation, we cannot exclude that the spontaneous activity may have masked RGC responses in our study – why this would have mostly affected 'Off' cells, however, is unclear.

## Differential degeneration of functional RGC types

How disease and degeneration (i.e. photoreceptor loss, glaucoma and optic nerve injury) affect RGC health and

function, in particular of specific types, has been the focus of an increasing number of studies in the past years (e.g. Care et al., 2019; Lee et al., 2022; Ou et al., 2016; Tran et al., 2019). Since they are morphologically easy to identify, $\alpha$-RGCs have been thoroughly studied. Generally, $\alpha$-cells are considered resilient to an insult such as optic nerve crush (ONC) (Duan et al., 2015). However, a recent ONC study found differences among $\alpha$-cells, with sustained $\alpha$-cells being more resilient than transient $\alpha$-cells (Tran et al., 2019). Similarly, in a transient ocular hypertension paradigm, a glaucoma model, 'Off-$\alpha$-transient' RGCs were the most vulnerable of the $\alpha$-types (Ou et al., 2016). In our study, we found all 'Off-$\alpha$' RGC ($G_{5,8}$) responses to be less abundant in *rd10 vs.* wild-type, whereas 'On-$\alpha$' RGC ($G_{19,24}$) responses persisted longer. This suggests that for photoreceptor loss as the insult, the divide is not between sustained and transient but rather between 'On-' and 'Off-$\alpha$' cells, with 'On-$\alpha$' cells being more resilient than their Off counterparts.

Generally, our data promote the idea that RGC resilience in photoreceptor degeneration depends on the retinal circuits involved (e.g. 'On'- *vs.* 'Off'-pathway): Already shortly after the onset of degeneration (P30), we found the fraction of 'Off'- and 'On–Off'-type RGCs to be reduced, whereas 'On'-RGCs remained functional until at least P45. Based on our data, the resilience of RGC response types can be ranked (from resilient to vulnerable): 'Uncertain' > 'Fast On' > 'Slow On' > 'On–Off' > 'Off'. Interestingly, this ranking is different from that for ONC, which was studied at the RGC type-level with transcriptomic data (Tran et al., 2019), and where, for instance, no differences between 'On'- and 'Off'-RGCs were detected.

The relative vulnerability we found in 'Off' cells was not seen in an earlier study of *rd10* mice, where 'On' and 'Off' responses were reported to be affected to a similar extent (Stasheff et al., 2011). Moreover, for a rat model of photoreceptor degeneration (P23H), it was reported that mainly 'Off' types remained responsive up to P300 (roughly P180 to P210 in mice), whereas 'On' and 'On–Off' responses sharply declined (Fransen et al., 2015). These differences may be related to the experimental approach (e.g. micro-electrode arrays *vs.* $Ca^{2+}$ imaging) or reflecting mutation- and/or species-dependent differences in retinal remodelling.

The difference between 'On' and 'Off' we observed is likely not directly due to changes in BC input, since in *rd10* 'Off'-CBCs retain their function, while 'On'-CBC function is altered (Puthussery et al., 2009). Instead, a recent *rd10* study using voltage-clamp recordings found an imbalance of inhibition and excitation between the 'On'- and the 'Off'-pathways (Carleton & Oesch, 2024). Specifically, 'On'-type RGCs received mostly excitation, while 'Off'-type RGCs received mostly inhibition. Carleton and Oesch (2024) also established that the

inhibition in the 'Off'-pathway derives from glycinergic ACs and is stronger in *rd10* than in wild-type retina. Interestingly, this is in line with previous work on ocular hypertension, where the 'Off'-sublamina of the IPL was demonstrated to be less 'stable', displaying signs of disorganisation earlier than the 'On'-sublamina (Ou et al., 2016). Moreover, the 'Off'-stratifying dendrites of 'On–Off'-RGCs lost their branching complexity more rapidly than the 'On'-stratifying dendrites, while the excitatory synapses onto both 'On' and 'Off' dendrites decrease similarly. Whether a comparable change in inner retinal synaptic organisation also happens in *rd10* remains to be tested in future studies.

The differential loss of 'Off'-type responses we observed in *rd10* is consistent with increased inhibition of the 'Off'-pathway, possibly involving synaptic reorganisation in the inner retina (as detailed above). However, other disease-triggered mechanisms likely also play a role. These mechanisms include changes in gene expression, neurotrophic factors and Müller glia activation. For example, that RBCs switch their synaptic inputs to cones may trigger new gene expression patterns of glutamate receptors (Marc et al., 2007). Furthermore, BCs rely on neurotrophic signals from photoreceptors to maintain their dendritic arbours; the absence of these factors in the *rd* condition leads to atrophy (Strettoi & Pignatelli, 2000; Strettoi et al., 2002). Differential gradients of neuro-trophic factors might also help preserve the functional layering of the IPL (Ou et al., 2016). Finally, Müller cell activation is significantly up-regulated shortly after rod loss peaks (Gargini et al., 2007). It is therefore conceivable that these glia cells become heavily involved in clearing rod remnants, which could reduce their capacity to regulate neurotransmitter uptake and maintain ion homeostasis. In turn, heightened metabolic demands and altered glial function could contribute to the distinct vulnerabilities observed in 'Off'-pathways, which were shown to be metabolically more demanding (Kageyama & Wong-Riley, 1984).

Taken together, we think that the finding of pre-dominantly functional super-group-specific effects on *rd10* RGCs argues for remodelling of the upstream circuits rather than changes in intrinsic properties. At the general level, our data in comparison with earlier studies, promote a view of RGC resilience being surprisingly heterogeneous and depending not only on the genetic or functional response type, but also on the type of insult to the tissue.

## Identifying cell types over the course of a progressive disease

To identify known functional RGC types, we utilised a published classifier (Gonschorek et al., 2024; Qiu et al., 2023) that matched our data to an established dataset

(Baden et al., 2016). Specifically, this approach allowed us to estimate the probability (confidence score, see Methods) with which a cell in our wild-type and *rd10* datasets belongs to a distinct RGC type and, hence, enabled a type-resolved analysis. This analysis suggests, first, that the functional RGC diversity in *rd10* is robust and comparable to that in wild-type at least until P90, and second, there is a differential, RGC-type-dependent loss in responsiveness. Still, this method has its limitations, which we discuss in the following (Fig. 10).

To examine if the classifier was biased towards 'On' over 'Off' cells, we computed an 'On–Off' index (OOi) for the MB responses, ignoring the classifier's type labels. Overall, we found fewer 'Off'-components in the *rd10* dataset, independent of age (Fig. 10*B*). This indicates that the *rd10* dataset indeed contains fewer 'Off' response components, supporting our results from the classifier analysis and our conclusion that 'Off'-types may be more vulnerable in *rd10*.

As the classifier was trained with responses from healthy wild-type retina, it cannot identify 'novel' response patterns (i.e. out-of-distribution data) that may develop during disease progression. As a consequence, such novel response types will be excluded from the analysis due to their low confidence score (CS). To examine this potential issue, we asked what percentage of cells were discarded by our CS threshold (CS $\geq$ 0.25) compared to having no CS threshold (Fig. 10*C*). We found for wild-type, while the percentage of classified cells decreased (as expected), it remained stable within each functional super-group. In *rd10*, however, the percentage of 'Off' and 'On–Off' cells was substantially reduced compared to wild-type, and their percentage decreased with age. This suggests that, indeed, the lower abundance of RGC types with 'Off' components may be explained by the classifier being less reliable in identifying their responses with progressing degeneration – either because their responses become too noisy or they changed their profile substantially.

In future studies it will be important to test the possibility of an RGC type significantly changing its response profile, for example, by combining functional and transcriptomic methods in the same tissue.

### *rd10* and vision restoration

To develop successful vision restoration approaches, a better knowledge of the inner retinal circuits during disease progression is crucial. Notably, different types of pathologies (e.g. RP, glaucoma) appear to lead to differential degeneration of the RGCs. Especially in a genetically highly heterogeneous disease like RP, where a general restoration strategy may build upon the relative stability of the inner retina, it is important to know which pathways are more resilient than others. Our data suggest that interventions, such as the expression of light-sensitive channels in BCs, may be more efficient in the 'On'-pathways, which turned out to be more resilient in *rd10*. Regarding the time point, in *rd10* mice P45 may be most suitable, when rods are mostly lost but functional cones still persist. This time point is roughly equivalent to that when most patients are diagnosed with RP: when their night vision is heavily impaired if not lost and daylight vision is still functional (Hamel, 2006; Hartong et al., 2006). At later time points, therapeutic approaches are likely to be severely hampered by increasing (functional) remodelling of inner retinal circuits.

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

## Additional information

### Data availability statement

Data as well as all custom analyses including code and notebooks to reproduce analyses is available at https://github.com/eulerlab/rd10_rgc and https://doi.org/10.12751/g-node.89z5px.

### Competing interests

The authors declare no conflicts of interests.

### Author contributions

T.S. and T.E. designed the study with input from N.D.; N.D. performed functional imaging experiments; M.H. and N.D. performed immunohistochemistry experiments with help from T.S.; N.D. performed pre-processing; D.G., J.O. and N.D. analysed the data with the help of Y.Q.; T.S. wrote the animal protocol, N.D. and D.G. wrote the manuscript with the help from T.E., T.S., J.O. and Y.Q. All authors were involved in data interpretation and editing the manuscript, approved the final version and agree to be accountable for all aspects of the work in ensuring that questions related to the accuracy or integrity of any part of the work are appropriately investigated and resolved. All persons designated as authors qualify for authorship, and all those who qualify for authorship are listed.

### Funding

We acknowledge support by the Open Access Publishing Fund of the University of Tübingen. This work was funded by the Tistou & Charlotte Kerstan Stiftung (RI-FG P3 EU/SCH 1–2 & 3 Dysz) and the German Research Foundation (DFG; SFB 1233 'Robust Vision', 276693517; EU 42/10-1; EU 42/12-1).

## Acknowledgements

We thank François Paquet-Durand, Simon Clark and Günter Zeck for lively and inspiring discussions, and Olga Oleksiuk for excellent microscopy support.

## Keywords

cell types, ganglion cells, photoreceptor degeneration, *rd10* mutant mouse, retina, retinal function, retinitis pigmentosa

## Supporting information

Additional supporting information can be found online in the Supporting Information section at the end of the HTML view of the article. Supporting information files available:

**Peer Review History**

