## [Peer Review History · The Journal of Physiology]

Photoreceptor degeneration has heterogeneous effects on functional retinal ganglion cell types

Thomas Euler, Nadine Dyszkant, Jonathan Oesterle, Yongrong Qiu, Merle Harrer, Timm Schubert, and Dominic Gonschorek
DOI: 10.1113/JP287643

Corresponding author(s): Thomas Euler (thomas.euler@cin.uni-tuebingen.de)

Review Timeline:	Submission Date:	11-Sep-2024
	Editorial Decision:	16-Oct-2024
	Revision Received:	17-Dec-2024
	Accepted:	17-Jan-2025

Senior Editor: Nathan Schoppa

Reviewing Editor: Jonathan Demb

Transaction Report:

Dear Dr Euler,

Re: JP-RP-2024-287643 "Photoreceptor degeneration has heterogeneous effects on functional retinal ganglion cell types" by Thomas Euler, Nadine Dyszkant, Jonathan Oesterle, Yongrong Qiu, Merle Harrer, Timm Schubert, and Dominic Gonschorek

Thank you for submitting your manuscript to The Journal of Physiology. It has been assessed by a Reviewing Editor and by 2 expert referees and we are pleased to tell you that it is potentially acceptable for publication following satisfactory major revision.

Please address all the points raised and incorporate all requested revisions or explain in your Response to Referees why a change has not been made. We hope you will find the comments helpful and that you will be able to return your revised manuscript within 2 months. If you require longer than this, please contact journal staff: jp@physoc.org. Please note that this letter does not constitute a guarantee for acceptance of your revised manuscript.

LANGUAGE EDITING AND SUPPORT FOR PUBLICATION: If you would like help with English language editing, or other article preparation support, Wiley Editing Services offers expert help, including English Language Editing, as well as translation, manuscript formatting, and figure formatting at www.wileyauthors.com/eoo/preparation. You can also find resources for Preparing Your Article for general guidance about writing and preparing your manuscript at www.wileyauthors.com/eoo/prepresources.

REVISION CHECKLIST:

We look forward to receiving your revised submission.

Yours sincerely,

Nathan Schoppa
Senior Editor
The Journal of Physiology

REQUIRED ITEMS

- Author photo and profile. First or joint first authors are asked to provide a short biography (no more than 100 words for one author or 150 words in total for joint first authors) and a portrait photograph. These should be uploaded and clearly labelled together in a Word document with the revised version of the manuscript. See Information for Authors for further details.

- You must start the Methods section with a paragraph headed Ethical approval (https://jp.msubmit.net/cgi-bin/main.plex?form_type=display_requirements#methods).

Research must comply with The Journal's policies regarding animal experiments (<https://physoc.onlinelibrary.wiley.com/hub/animal-experiments>) and adherence to these policies must be stated in the manuscript.

Authors should confirm in their Methods section that their experiments were carried out according to the guidelines laid down by their institution's animal welfare committee, including an ethics approval reference number. The Methods section must contain a statement about access to food, water and housing, details of the anaesthetic regime: anaesthetic used, dose and route of administration, and method of killing the experimental animals.

- The reference list must be in alphabetical order, rather than numbered, to comply with our Journal format.

- Your manuscript must include a complete Additional Information section, including competing interests; funding; author contributions and acknowledgements.

- Please upload separate high-quality figure files via the submission form.

- Please ensure that any tables are editable and in Word format, and wherever possible, embedded in the article file itself.

- Please ensure that the Article File you upload is a Word file.

- Your paper contains Supporting Information of a type that we no longer publish, including supplementary tables and

figures. Any information essential to an understanding of the paper must be included as part of the main manuscript and figures. The only Supporting Information that we publish are video and audio, 3D structures, program codes and large data files. Your revised paper will be returned to you if it does not adhere to our Supporting Information Guidelines.

- Papers must comply with the Statistics Policy: https://jp.msubmit.net/cgi-bin/main.plex?form_type=display_requirements#statistics.

In summary:

- If $n \leq 30$, all data points must be plotted in the figure in a way that reveals their range and distribution. A bar graph with data points overlaid, a box and whisker plot or a violin plot (preferably with data points included) are acceptable formats.

- If $n > 30$, then the entire raw dataset must be made available either as supporting information, or hosted on a not-for-profit repository, e.g. FigShare, with access details provided in the manuscript.

- 'n' clearly defined (e.g. x cells from y slices in z animals) in the Methods. Authors should be mindful of pseudoreplication.

- All relevant 'n' values must be clearly stated in the main text, figures and tables.

- The most appropriate summary statistic (e.g. mean or median and standard deviation) must be used. Standard Error of the Mean (SEM) alone is not permitted.

- Exact p values must be stated. Authors must not use 'greater than' or 'less than'. Exact p values must be stated to three significant figures even when 'no statistical significance' is claimed.

- A Data Availability Statement is required for all papers reporting original data. This must be in the Additional Information section of the manuscript itself. It must have the paragraph heading 'Data Availability Statement'. All data supporting the results in the paper must be either: in the paper itself; uploaded as Supporting Information for Online Publication; or archived in an appropriate public repository. The statement needs to describe the availability or the absence of shared data. Authors must include in their statement: a link to the repository they have used, or a statement that it is available as Supporting Information; reference the data in the appropriate sections(s) of their manuscript; and cite the data they have shared in the References section. Whenever possible, the scripts and other artefacts used to generate the analyses presented in the paper should also be publicly archived. If sharing data compromises ethical standards or legal requirements then authors are not expected to share it, but must note this in their statement. For more information, see our Statistics Policy.

- Please include an Abstract Figure file, as well as the Figure Legend text within the main article file. The Abstract Figure is a piece of artwork designed to give readers an immediate understanding of the research and should summarise the main conclusions. If possible, the image should be easily 'readable' from left to right or top to bottom. It should show the physiological relevance of the manuscript so readers can assess the importance and content of its findings. Abstract Figures should not merely recapitulate other figures in the manuscript. Please try to keep the diagram as simple as possible and without superfluous information that may distract from the main conclusion(s). Abstract Figures must be provided by authors no later than the revised manuscript stage and should be uploaded as a separate file during online submission labelled as File Type 'Abstract Figure'. Please also ensure that you include the figure legend in the main article file. All Abstract Figures should be created using BioRender. Authors should use The Journal's premium BioRender account to export high-resolution images. Details on how to use and access the premium account are included as part of this email.

EDITOR COMMENTS

Reviewing Editor:

This study uses two-photon calcium imaging to assess retinal function following photoreceptor degeneration. The authors examine a mouse model of retinitis pigmentosa and measure how different types of retinal ganglion cell (RGC) react to the slow loss of cone photoreceptor function that follows the sharp decline in rod function.

The methods entail quantitative characterizations of calcium imaging to visual stimulus protocols designed to distinguish ~40 types of RGC. The cell-typing depends on the functional readout of the calcium signal, and thus it could become difficult to track a specific RGC type in a diseased retina, since certain types will lose their characteristic response. With that said, there is clearly a larger defect in the responses of RGC types with OFF responses, whereas RGCs with ON responses continue to respond at later stages of disease.

The reviews raise several concerns, some of which could be addressed with additional experiments. For example, spike recordings could be used as a complementary method to test whether OFF RGCs lose responses at later stages of disease - ruling out an alternative explanation that this finding relates to an artifact associated with saturation of the calcium signal caused by elevated spontaneous activity (related to Reviewer 1's comments).

It would be difficult to distinguish rod- vs. cone-mediated responses with two-photon imaging, since the laser likely saturates the rod responses (related to Reviewer 2's comments).

Most of the specific concerns seem addressable with changes to the text and additional analysis.

Please also see 'Required Items' above.

Senior Editor:

Comments for Authors to ensure the paper complies with the Statistics Policy (Required):

N-values are not consistently given in the Results or Figures. There appears to be information about N-values in the Methods but it is much too difficult to evaluate the N-values associated with each statistical comparison. N-values should be in the Results and provided for each comparison. Also, if $N < 30$, plots in the figures should show individual data points.

Comments to the Author:

Your manuscript has been evaluated by two expert reviewers and a reviewing editor, and has been considered of interest and potentially suitable for publication. However, a number of major concerns were raised, which would need to be addressed, and a revised manuscript re-reviewed. An ultimate decision about acceptance will depend on your addressing the concerns in a satisfactory manner.

Amongst the most important concerns that will require additional experiments and analysis include the need for control experiments to test for elevated baseline activity and, also, questions about whether cells can be accurately classified based on their functional properties when those are changing in the disease model. Reviewer 2 suggests comparing rod versus cone-mediated activity at different time-points, but, as pointed out by the reviewing editor, this may be difficult to assess using two-photon imaging and the concern better addressed with some discussion. All concerns that were raised would need to be addressed.

I would like to raise a number of additional points that will require changes:

(1) N-values are not consistently given in the Results or Figures. There appears to be information about N-values in the Methods but it is much too difficult to evaluate the N-values associated with each statistical comparison. N-values should be in the Results and provided for each comparison. Also, if $N < 30$, plots in all of the figures should show individual data points.

(2) Around animal use, the authors will need to describe the animals' access to food and water.

(3) The authors include Supplementary Figures. While such figures may be used in the manuscript review process to show less essential results, the Journal of Physiology has a policy of not allowing Supplementary Figures for final publication. The authors will need to determine which, if any, of the Supplementary Figures will need to be included as regular figures.

(4) The authors will need to follow all formatting conventions for Journal of Physiology, for example having the Methods immediately after the Introduction.

REFeree COMMENTS

Referee #1:

Tracking the progression of retinal degenerations (RDs), such as Retinitis Pigmentosa (RP), is crucial for developing vision restoration strategies. In this study, Euler and colleagues examined changes in the activity of different retinal ganglion cell (RGC) types in the rd10 mouse model that simulates the slow progression of human RP. They employed high-throughput two-photon calcium imaging to characterize functional RGC types in both wild-type and rd10 mice across three to four ages. Their findings revealed that RGC soma density and cell types in the rd10 mice were comparable to those in wild-type mice at various postnatal ages; however, the proportion of light-responsive cells decreased as the disease progressed, starting from P45. By P180, no light-responsive RGCs were detected. Additionally, they observed slight age-related changes in the spatial and temporal receptive field properties. The study's primary conclusion is that OFF and ON-OFF RGCs are more vulnerable to degeneration than ON RGC types. Overall, these results contribute to the growing body of research on functional alterations in retinal cell types during degeneration.

Overall, the experiments are rigorously conducted and adhere to the protocols established in previous RGC classification studies by this group. However, the primary conclusion that OFF RGCs degenerate more rapidly in rd10 mice is compromised by a technical limitation. OFF cells are likely to exhibit higher spontaneous activity (2-10 Hz), which could lead to saturation of the Ca²⁺ indicator and potentially obscure OFF RGC responses. The authors should include a control experiment to address this concern, such as applying NBQX/AP5 to demonstrate that baseline activity remains unaffected.

A second significant concern relates to the clustering and correlation analysis. If variations in RGC responses are used to categorize cells into different types, it is unclear how reliable the classification can be when response profiles change during the progression of degeneration. This raises questions about the stability of cell-type classification in the context of a degenerating retinal environment. Finally, I'm left wondering what was the point of this massive effort, when the end conclusion requires a simple ON, ON-OFF and OFF classification scheme.

Other comments

1. The degeneration pattern in the rd10 mouse follows a center-to-periphery gradient, which should be taken into account for activity comparisons in their analysis.
2. Lines 89-90: The authors should reference prior studies that document changes in synaptic connectivity in the outer and inner retina of rd10 mice (Phillips et al., 2010; Puthussery et al., 2009; Barhoum et al., 2008; Gargini et al., 2007) as well as the generation of spontaneous activity in this mouse model (Goo et al., 2011; Stasheff et al., 2011; Biswas et al., 2014; Toychiev et al., 2013; Haselier et al., 2017).
3. A substantial body of research has reported spontaneous activity/oscillations in various RP models (Haq et al., 2014; Goo et al., 2011; Stasheff et al., 2011; Biswas et al., 2014; Toychiev et al., 2013; Haselier et al., 2017; Borowska et al., 2011; Trenholm et al., 2012; Choi et al., 2014; Margolis et al., 2008; Tu et al., 2015; Menzler and Zeck, 2011). This activity has been shown to impact the signal-to-noise ratio of evoked responses (Yee et al., 2012) and interfere with the effectiveness of electrical stimulation (Haselier et al., 2017). Additionally, reducing this aberrant activity has been found to restore light responses (Toychiev et al., 2013; Barrett et al., 2016). The authors should address why similar activity was not observed during the later stages of degeneration in their study. Could the low percentage of responding cells at P90 and P180 be attributed to the inhibitory effect of spontaneous activity on the OFF pathway?
4. Figure 3: Please include scale bars. Additionally, do the authors observe any differences in the amplitude of responses between wild-type and rd10 mice?
5. Figure 5: It would be informative to include examples where the correlation between wild-type and rd10 responses is low and changes over the course of disease progression, such as in G2, G16, and G26 (Figure S5).

6. Lines 343-348: The main peak lag values/p-values mentioned in the text for G17 do not correspond with the values displayed in the plot (Figure 6C).
7. Figure 6: Given the cell-to-cell variability in the mean peak lag, interpreting the population data is challenging. Additionally, there is an error in the legend regarding significance values; values < 0.05 should be denoted by '***' instead of '**'.
8. Line 402: Please correct the order of RGC vulnerability.
9. Figure S3: This figure seems redundant and could be summarized in the text. Additionally, Figure S7 could be combined with Figure S2 for clarity.

Referee #2:

Dyszkant et al. investigate changes to retinal ganglion cell response properties during photoreceptor degeneration caused by mutation in *pde6b*, the *rd10* mouse model. Specifically they looked at changes to distinct RGC types using 2-photon calcium imaging in response to chirp and grating stimulus. They found fewer OFF types, and found RFs trended toward being smaller and faster throughout degeneration, though largely type and timepoint dependent.

In general, I appreciated the authors' relation to work in the field and how this information is relevant to correcting retinitis pigmentosa. Good section on the limitations and pointing out different models are showing different things. The figures are beautifully made, illustrative of the points being made, and align with the text.

Corrections:

This paper would greatly benefit from sorting rod-mediated activity from cone-mediated activity, given the nature of cell loss (rod followed by cone). While light level is mentioned in methods (mesopic), this information is essential to include in the results section so the reader knows what pathway is being stimulated.

There are known changes in sensitivity at both the rod and cone level, and those vary depending on the timepoint. Isolating rod vs cone might also show interesting differences in RGC type-specific responses, ex. higher RGC dropout in rod-isolated path, or similarities due to rod and cone pathway integration, or cone path might have a different threshold than WT, or interesting compensations.... Etc.

The authors report cell density is not interrupted, which aligns with previous findings. Are functional mosaics interrupted? This would appear as holes in mosaics where RGC responses are dropping out from loss of upstream PRs. Please show representative mosaic of an impacted RGC type.

of genes causing RP is incorrect (references are 20+ years old & prior to completion of the Human Genome Project). Retnet says genes and loci for all types of RP are 88. Similarly, references for animal model diversity are 20-30 years old and are probably not the most reflective of where the field stands in 2024.

Are RGC types more vulnerable from their inherent cell biology or is upstream circuitry impacted, leading to type specific RGC changes? Please clarify text to reflect the changes are from impacted upstream changes. Unless that is not an accurate conclusion, in which case, please justify.

It would add clarity for the reader to add in % changes in the text for figure 6.

Missing refs/discussion points:

It would be beneficial to include discussion on how these models/methods (below) differ and what that tells us about the system.

*Sekirnjak 2011 P23H rat MEA of RGCs: RF size decreased, RF time increased, ON had decreased firing rate.

*Yu 2017 S334ter-3 rat, MEA of RGCs: looked at cone pathway and found altered temporal RFs, impaired DS. Change in spatial RF was proportional to size of WT RF (bigger shrunk more). This paper is mentioned in text (ref 48) but hard to compare data due to the scaling in figure 6. Are there correlations to size or speed as seen by Yu et al.?

*Fransen et al. 2015 P23H rat: sharp decline in ON and ON-OFF RGCs, OFF preserved at P300.

*Stasheff et al. 2011 rd1 and rd10 (going to ignore rd1): ON and OFF impacted to a similar degree in rd10. More sustained responses (maybe).

Line 402, the last 'On' should be 'Off'

Missing information related to gain and signal-to-noise, which have both been implicated in prior studies about RGC responses during retinitis pigmentosa. Are there cell type specific changes to either? Please include.

END OF COMMENTS

EDITOR COMMENTS

Reviewing Editor:

This study uses two-photon calcium imaging to assess retinal function following photoreceptor degeneration. The authors examine a mouse model of retinitis pigmentosa and measure how different types of retinal ganglion cell (RGC) react to the slow loss of cone photoreceptor function that follows the sharp decline in rod function.

The methods entail quantitative characterizations of calcium imaging to visual stimulus protocols designed to distinguish ~40 types of RGC. The cell-typing depends on the functional readout of the calcium signal, and thus it could become difficult to track a specific RGC type in a diseased retina, since certain types will lose their characteristic response. With that said, there is clearly a larger defect in the responses of RGC types with OFF responses, whereas RGCs with ON responses continue to respond at later stages of disease.

We thank the reviewing editor for appreciating our study. Based on the input by editors and reviewers, we performed additional experiments and analyses and revised our manuscript accordingly. For details, please see our replies below.

The reviews raise several concerns, some of which could be addressed with additional experiments. For example, spike recordings could be used as a complementary method to test whether OFF RGCs lose responses at later stages of disease - ruling out an alternative explanation that this finding relates to an artifact associated with saturation of the calcium signal caused by elevated spontaneous activity (related to Reviewer 1's comments).

We have addressed the concern regarding the 'Off' RGCs by performing additional experiments as suggested by reviewer #1 – for details and results, please see below.

It would be difficult to distinguish rod- vs. cone-mediated responses with two-photon imaging, since the laser likely saturates the rod responses (related to Reviewer 2's comments).

Yes, the excitation laser wavelength used excites the photoreceptors through several mechanisms (see Euler et al. Eur J Physiol 2009), which makes it challenging to reliably separate rod and cone signals. In the revised manuscript, we address this limitation now briefly in the Results and the Discussion.

Most of the specific concerns seem addressable with changes to the text and additional analysis.

Please also see 'Required Items' above.

Senior Editor:

Comments for Authors to ensure the paper complies with the Statistics Policy (Required):

N-values are not consistently given in the Results or Figures. There appears to be information about N-values in the Methods but it is much too difficult to evaluate the N-values associated with each statistical comparison. N-values should be in the Results and provided for each comparison. Also, if $N < 30$, plots in the figures should show individual data points.

We changed the presentation of the results such that it adheres to the statistics policy of the journal. We also added the n-values to the Figure Legends.

Comments to the Author:

Your manuscript has been evaluated by two expert reviewers and a reviewing editor, and has been considered of interest and potentially suitable for publication. However, a number of major concerns were raised, which would need to be addressed, and a revised manuscript re-reviewed. An ultimate decision about acceptance will depend on your addressing the concerns in a satisfactory manner.

Amongst the most important concerns that will require additional experiments and analysis include the need for control experiments to test for elevated baseline activity and, also, questions about whether cells can be accurately classified based on their functional properties when those are changing in the disease model. Reviewer 2 suggests comparing rod versus code-mediated activity at different time-points, but, as pointed out by the reviewing editor, this may be difficult to assess using two-photon imaging and the concern better addressed with some discussion. All concerns that were raised would need to be addressed.

We thank the reviewing editor for appreciating our study and for their input. Based on the input by editors and reviewers, we performed additional experiments and analyses and revised our manuscript accordingly. For details, please see our replies below.

I would like to raise a number of additional points that will require changes:

(1) N-values are not consistently given in the Results or Figures. There appears to be information about N-values in the Methods but it is much too difficult to evaluate the N-values associated with each statistical comparison. N-values should be in the Results and provided for each comparison. Also, if $N < 30$, plots in all of the figures should show individual data points.

We changed the presentation of the results such that it adheres to the statistics policy of the journal. We also added the n-values to the figure legends.

(2) Around animal use, the authors will need to describe the animals' access to food and water.

Done.

(3) The authors include Supplementary Figures. While such figures may be used in the manuscript review process to show less essential results, the Journal of Physiology has a policy of not allowing Supplementary Figures for final publication. The authors will need to determine which, if any, of the Supplementary Figures will need to be included as regular figures.

To avoid Supplementary Figures, we made the following changes:

Fig. 1: unchanged

Fig. 2: unchanged

Fig. 3: we added some of our IHC stainings from former S1

Fig. 4: unchanged

Fig. 5: new figure, combining panels from former S2, S4, and S7

Fig. 6: contains panels from former S6 (JSDs of QI distributions)

Fig. 7: former Fig. 5 (classification of RGC types and their correlation), with additional examples as requested by the reviewer

Fig. 8: former Fig. 6

Fig. 9: former Fig. 7 (abundance of cell types)

Fig. 10: new figure (classifier robustness with panels from former S5)

The remaining Supplementary Figure panels were discarded.

(4) The authors will need to follow all formatting conventions for Journal of Physiology, for example having the Methods immediately after the Introduction.

In the revised manuscript, we moved the Methods.

REFEREE COMMENTS

Referee #1:

Tracking the progression of retinal degenerations (RDs), such as Retinitis Pigmentosa (RP), is crucial for developing vision restoration strategies. In this study, Euler and colleagues examined changes in the activity of different retinal ganglion cell (RGC) types in the *r10* mouse model that simulates the slow progression of human RP. They employed high-throughput two-photon calcium imaging to characterize functional RGC types in both wild-type and *rd10* mice across three to four ages. Their findings revealed that RGC soma density and cell types in the *rd10* mice were comparable to those in wild-type mice at various postnatal ages; however, the proportion of light-responsive cells decreased as the disease progressed, starting from P45. By P180, no light-responsive RGCs were detected. Additionally, they observed slight age-related changes in the spatial and temporal receptive field properties. The study's primary conclusion is that OFF and ON-OFF RGCs are more vulnerable to degeneration than ON RGC types. Overall, these results contribute to the growing body of research on functional alterations in retinal cell types during degeneration.

We thank the referee for appreciating our study and for their helpful comments.

Overall, the experiments are rigorously conducted and adhere to the protocols established in previous RGC classification studies by this group. However, the primary conclusion that OFF RGCs degenerate more rapidly in *rd10* mice is compromised by a technical limitation. OFF cells are likely to exhibit higher spontaneous activity (2-10 Hz), which could lead to saturation of the Ca²⁺ indicator and potentially obscure OFF RGC responses. The authors should include a control experiment to address this concern, such as applying NBQX/AP5 to demonstrate that baseline activity remains unaffected.

Following the reviewer's suggestion, we performed additional experiments.

Specifically, we recorded from *rd10* mouse retinæ at P90, presented visual stimuli and then bath-applied a mixture of D-AP5 (50 µM) and CNQX (25 µM). The recordings largely followed the experimental paradigm described in the manuscript, except that we mounted and electroporated each retina wing (= quarter) separately and only recorded one field per wing. This was done to keep a consistent timing between the control (before washing in the drugs) and the drug recordings. Also, we did not attempt washing out the drugs. The adapted paradigm was as follows:

First, we played the chirp and moving-bar (MB) stimuli (as for all our data) to allow us to identify RGC types, if needed. Next, we recorded for 2 minutes without playing a stimulus (baseline) and then presented a MB-break stimulus, a series of 4 MB stimuli (8 directions, 1 repeat) separated by 1-minute breaks, resulting in an 8-minutes long sequence. After the first stimulus sequence of the MB-break stimulus, the drug mixture was washed in.

We first tested this paradigm using a wild-type (wt) retina without drugs to see if the responses remained stable during such a long recording. Since this was the case, we went on and performed the drug experiments on *rd10* retinæ. In total, we analysed 4 *rd10* retinæ (n=3 mice with n=8 fields in total).

We extracted ROIs and the respective traces as described in the manuscript, but, because of the long recording time, we applied an additional motion correction step to

the image stack to reduce the effects of small tissue drifts on the tissue (Pnevmatikakis and Giovannucci, J Neurosci Meth 2017).

We computed the pre-drug and post-drug calcium baselines for the MB-break stimulus as the median of the trace in 30 second windows; the former just before the first MB sequence, the latter 1 minute after the last MB sequence. The baseline change was computed as the post-drug baseline minus the pre-drug baseline. This difference was normalised by dividing by the standard deviation of the pre-drug baseline.

We found drug effects to be visible quite quickly, that is, approx. 3 minutes after wash-in, the light responses were gone. Moreover, we found that the baseline was reduced on averages after the drug application (see Fig. R1A below). This change was similar in both On- and Off-RGCs, arguing against a spontaneous activity-related effect specific to Off-RGCs. In addition, the D-AP5/CNQX-induced baseline shift was not significantly correlated with response quality (Fig. R1B), suggesting no direct link between spontaneous activity and baseline changes.

These results suggest that in *rd10* mice, spontaneous activity, which has been shown to result largely from network activity (Goo et al. Korean J. Physiol. Pharmacol. 2011, Trenholm et al. J. Physiol. 2012, Biswas et al PLOS ONE 2014, Choi et al. J. Neurophysiol. 2014), affects On and Off RGCs similarly – at least at P90. Of course, more complex (synaptic) interactions are conceivable, because blocking ionotropic glutamate receptors does not only remove excitatory input from RGCs but also from bipolar cells and amacrine cells. Still, our data suggest that the calcium baseline is similar in ‘On’ and ‘Off’ RGCs.

Figure R1. Effects of D-AP5 + CNQX on the baseline Ca^{2+} levels. (A) Drug-induced baseline changes as a function of Off-On index (OOi). Data points represent cells. Linear regression line is shown with bootstrapped 95% confidence intervals. Pearson’s r between the On-Off index and the baseline shift was not significant. (B) Baseline changes as a function of MB response quality index (Ql_{MB}). Data points represent cells and are colour-coded by OOi. Linear regression and Pearson’s r were computed for four groups: “all” (black): all cells, “On” (blue): all cells with an OOi $> +0.2$, “Off” (red): all cells with an OOi < -0.2 , and “Off” (orange): all cells that were assigned to an Off group by the classifier (no quality or CS filters were applied here).

However, because these are difficult to interpret, we decided not to add the data to our revised manuscript. However, we would be happy to do so, if requested by the reviewer.

A second significant concern relates to the clustering and correlation analysis. If variations in RGC responses are used to categorize cells into different types, it is unclear how reliable the classification can be when response profiles change during the progression of degeneration. This raises questions about the stability of cell-type classification in the context of a degenerating retinal environment. Finally, I'm left wondering what was the point of this massive effort, when the end conclusion requires a simple ON, ON-OFF and OFF classification scheme.

Thank you for raising this point – in fact, we were discussing this question quite a lot and we appreciate the opportunity to elaborate.

First of all, we agree that a supervised learning classifier, such as a Random Forest Classifier, is limited to confidently classifying samples (=assigning labels) from the data distribution it was initially trained on. Hence, if an RGC type's response profile changes (e.g., as a consequence of the photoreceptor degeneration and the subsequent change in input), this will affect the classifier's reliability. How large this effect on the classifier performance is depends on the response changes: Overall noisier responses are likely to have less of an effect than a shift in data distribution. Therefore, we initially also tested clustering approaches: These are "open" to new response types, which then, e.g., end up in clusters that are only found in *rd10* but not wt. However, we did not find clear evidence for *rd10*-exclusive clusters in this preliminary clustering analysis.

In contrast, the classifier uses an ensemble of decision trees, based on which it provides a confidence score for each label (here RGC type), indicating how confident this classification is. If a confidence score (CS) is too low, the cell's response has deteriorated from a healthy, functional response type; such a cell was taken out of the further analysis. This approach allowed us to ask, how "stable" the RGC response types in *rd10* over the course of the degeneration were, and this is why we opted for the classifier.

Let us consider the most **likely three scenarios that can occur during degeneration**: If the response profile of an RGC (1) "just" got noisier, its CS would decrease and below a defined threshold (<0.25), the cell is discarded. However, if the response profile changed in some systematic way, it could either (2) change such that it does not match any type the classifier was trained on or (3) appear like that of a different RGC type.

To address **case (1)**, we computed the overall standard deviation (SD) of the responses to chirp and moving bar in wt and *rd10* mice for each age. Between ages and mouse lines, the SD values were fairly similar (Fig. R2A). Then, we used the *rd10* SD for each age and added 25% or 50% Gaussian noise with this SD to the wt response traces to "simulate" biologically plausible noise (Fig. R2B). Next, we used the classifier to predict the cell type labels based on the noisy traces and analysed how many cells changed their type assignments depending on the noise level. For this, we compared their type label in the "no noise" with the "25% noise" or the "50% noise" conditions, respectively (Fig. R2C). Overall, we found the classifier to be reliable in its type assignments as its prediction with 25% and 50% noise were still high for all ages. This suggests that noise

– as it may result from less light-driven input with progressing photoreceptor degeneration – could lead to lower CS (and more discarded cells) but not to substantial misclassifications.

Figure R2. Effects of noise of classifier performance. (A) grey: wild-type, magenta: *rd10*. For details, see text.

For **case (2)**, the classifier will try to assign the novel (out-of-the-distribution) response profile to different RGC types, which typically will result in a low CS and the cell is discarded. We computed the ratio of classified cells between $CS > 0$ (i.e. no threshold; see new Fig. 10C) and $CS \geq 0.25$ (as used in the manuscript). Here, we found that when applying $CS \geq 0.25$, the ratios of classified cells progressively decreased exclusively for Off and On-Off types in *rd10*. Because this effect was not seen in the wt, the loss of Off-components may indeed be attributed to a change in response profile (or an increase in noise, see **case (1)**).

For **case (3)**, the classifier would provide a high CS and assign the cell to a different RGC type. While we cannot track the fate of individual cells, we may detect this – if it happens systematically – as the abundance of the first RGC type would decrease and that of the second RGC type increase. We expect most of these changes to occur within the functional super-groups, therefore only having a limited influence on the conclusions of our study.

To investigate **if the classifier might be biased towards On- over Off-components**, we computed an On-Off-Index based on the MB responses – ignoring the classifier’s type label. Here, we compared the distributions of Off, On-Off, and On-components between ages and mouse lines, asking if the classifier underestimates the abundance of Off-types or if there were really fewer Off-types in the *rd10* vs. wt datasets. To this end, we binned the distribution into 3 bins (Off: -1 to -0.2; On-Off: -0.2 to 0.2; On: 0.2 to 1). Overall, we found fewer Off-components in the entire *rd10* dataset, that is, independent of age (new Fig. 10B). This supports that the *rd10* dataset indeed contains fewer Off response components, supporting our results for the classifier analysis and our conclusion that Off-types may be more vulnerable in *rd10*.

These considerations are reflected in the revised manuscript by the addition of new Fig. 10 and respective text in the Discussion.

Finally, in retrospect, our analysis seems like a large effort, but we did not know nor expect that the main effects in *rd10* would at the level of the super-groups (e.g., largely ‘Off’ and ‘On-Off’ cells vs. ‘Fast/Slow On’ and ‘Uncertain’). Moreover, we did see changes at the cell type level and now give examples in the revised manuscript.

Other comments

1. The degeneration pattern in the *rd10* mouse follows a center-to-periphery gradient, which should be taken into account for activity comparisons in their analysis.

Thank you; this is a good point. Indeed, we accounted for that by recording at locations across the retina (cf. Fig. 2B). To roughly estimate the effect of such a centre-to-periphery gradient of degeneration, we utilized the quality indices (QI) for chirp and MB stimuli as proxy of RGC responsiveness. Indeed, we found generally lower QI values in *rd10* in the central vs. the peripheral retina (centre vs. periphery: $QI_{\text{chirp}} = 0.256$ vs. 0.312 ; $QI_{\text{MB}} = 0.509$ vs. 0.55), however, the same trend was also present in wt (centre vs. periphery: $QI_{\text{chirp}} = 0.324$ vs. 0.386 , $QI_{\text{MB}} = 0.606$ vs. 0.662). Therefore, in our data, we cannot clearly observe a *rd10*-specific gradient. We think that this is likely due to the selection of recording fields, which practically was guided by the structural quality and general responsiveness of the area, resulting in a biased sampling.

We added a few sentences regarding this point to the revised manuscript.

2. Lines 89-90: The authors should reference prior studies that document changes in synaptic connectivity in the outer and inner retina of *rd10* mice (Phillips et al., 2010; Puthussery et al., 2009; Barhoum et al., 2008; Gargini et al., 2007) as well as the generation of spontaneous activity in this mouse model (Goo et al., 2011; Stasheff et al., 2011; Biswas et al., 2014; Toychiev et al., 2013; Haselier et al., 2017).

Thank you – the missing references are now cited in the revised manuscript.

3. A substantial body of research has reported spontaneous activity/oscillations in various RP models (Haq et al., 2014; Goo et al., 2011; Stasheff et al., 2011; Biswas et al., 2014; Toychiev et al., 2013; Haselier et al., 2017; Borowska et al., 2011; Trenholm et al., 2012; Choi et al., 2014; Margolis et al., 2008; Tu et al., 2015; Menzler and Zeck, 2011). This activity has been shown to impact the signal-to-noise ratio of evoked responses (Yee et al., 2012) and interfere with the effectiveness of electrical stimulation (Haselier et al., 2017). Additionally, reducing this aberrant activity has been found to restore light responses (Toychiev et al., 2013; Barrett et al., 2016). The authors should address why similar activity was not observed during the later stages of degeneration in their study. Could the low percentage of responding cells at P90 and P180 be attributed to the inhibitory effect of spontaneous activity on the OFF pathway?

We thank the reviewer for raising this important point. Initially, we were surprised by the lack of spontaneous activity, especially at later time points. But as this activity is usually rhythmic and has a frequency around 10 Hz (as reported by, e.g., Margolis et al. J. Neurosci. 2008, Goo et al. Korean J. Physiol. Pharmacol. 2011), which we cannot resolve it at our scan rate of 7.8125 Hz.

We now mention this limitation of our method in the Results and added a short paragraph on spontaneous activity in *rd10* to the Discussion.

4. Figure 3: Please include scale bars. Additionally, do the authors observe any differences in the amplitude of responses between wild-type and rd10 mice?

Sorry for the missing scale bars; they have now been added.

We agree that comparing absolute response amplitudes between wt and *rd10* would be interesting. However, this not possible because the Ca²⁺ indicator does not enable ratiometric measurements. The Ca²⁺ traces we show are normalized (by z-scoring; see Methods for details) and carry no information about absolute (resting) Ca²⁺ levels.

5. Figure 5: It would be informative to include examples where the correlation between wild-type and rd10 responses is low and changes over the course of disease progression, such as in G2, G16, and G26 (Figure S5).

Thanks for this suggestion; we have now added more examples to the figure (now Fig. 7).

6. Lines 343-348: The main peak lag values/p-values mentioned in the text for G17 do not correspond with the values displayed in the plot (Figure 6C).

Sorry about that mistake – we corrected the corresponding p-value (0.082 -> 0.0082).

7. Figure 6: Given the cell-to-cell variability in the mean peak lag, interpreting the population data is challenging. Additionally, there is an error in the legend regarding significance values; values < 0.05 should be denoted by '***' instead of '*'.

We see the reviewer's point and have revised the manuscript to address this issue. Specifically, we have softened our conclusions in the Results to acknowledge the challenges in interpreting population-level trends due to this variability. Additionally, we have added two sentences to the conclusion of the RF section.

Thank you also for pointing out the mistake in the significance values. We changed “* < 0.01” to “*** < 0.01” in the revised manuscript.

8. Line 402: Please correct the order of RGC vulnerability.

Corrected.

9. Figure S3: This figure seems redundant and could be summarized in the text. Additionally, Figure S7 could be combined with Figure S2 for clarity.

As the journal does not allow Supplementary Figures/Material, we reorganised the figures (for details, see our reply to the Editors' comments).

Referee #2:

Dyzskant et al. investigate changes to retinal ganglion cell response properties during photoreceptor degeneration caused by mutation in *pde6b*, the *rd10* mouse model. Specifically they looked at changes to distinct RGC types using 2-photon calcium imaging in response to chirp and grating stimulus. They found fewer OFF types, and found RFs trended toward being smaller and faster throughout degeneration, though largely type and timepoint dependent.

In general, I appreciated the authors' relation to work in the field and how this information is relevant to correcting retinitis pigmentosa. Good section on the limitations and pointing out

different models are showing different things. The figures are beautifully made, illustrative of the points being made, and align with the text.

We thank the reviewer for appreciating our study and for their helpful comments.

Corrections:

This paper would greatly benefit from sorting rod-mediated activity from cone-mediated activity, given the nature of cell loss (rod followed by cone). While light level is mentioned in methods (mesopic), this information is essential to include in the results section so the reader knows what pathway is being stimulated.

Thank you for your suggestion. In fact, our light conditions were probably in the low photopic range. We added this information to the beginning of the Results.

There are known changes in sensitivity at both the rod and cone level, and those vary depending on the timepoint. Isolating rod vs cone might also show interesting differences in RGC type-specific responses, ex. higher RGC dropout in rod-isolated path, or similarities due to rod and cone pathway integration, or cone path might have a different threshold than WT, or interesting compensations.... Etc.

We thank the reviewer for these valuable comments.

With respect to the light levels: We conducted our experiments under low photopic light levels, where responses are expected to be dominated by cone pathways. The infrared 2P excitation laser drives photoreceptors through different mechanisms (for details, see Euler et al. Eur J Physiol 2009; Euler et al., Multiphoton Microscopy, Neuromethods 2019) and thus adds a background stimulus independent of the actual light stimulus. We estimated the photoisomerization rates caused by the combined illumination (laser + stimulus) to range in the low photopic range.

Because we already operate in the low photopic regime, isolating rod- vs cone-driven responses is challenging. In an earlier study (Szatko et al. Nat Commun 2020), we were able to demonstrate rod-mediated responses with 2P imaging, but this was in the wt retina. In *rd10*, the stimuli are expected to predominantly drive cones, simply because rod degeneration is already extensive at P30 (Barhoum et al. Neuroscience 2008).

Taken together, while we agree with the reviewer that studying rod-cone pathway interactions during progressive photoreceptor degeneration would be interesting, our experimental approach is not quite suitable for this. To clarify this limitation, we added a few sentences to the Results of the revised manuscript.

The authors report cell density is not interrupted, which aligns with previous findings. Are functional mosaics interrupted? This would appear as holes in mosaics where RGC responses are dropping out from loss of upstream PRs. Please show representative mosaic of an impacted RGC type.

We agree with the reviewer that analysing the RGC mosaics would be instructive. Our recording method allows us to visualize all somata in the GCL and, hence, we can analyse the overall cell density directly. However, resolving functional RGC mosaics in a way that we can detect gaps is not feasible with our current dataset (see details below). For this, much larger fields would be needed, but this would have been at the cost of signal quality, because the time to collect fluorescence per somata (at the same

frame rate) would have been shorter. Because we expected lower response quality in *rd10*, we did not want to compromise with respect to signal-to-noise.

With the selected recording field size, we can record ~50-60 RGCs (considering that ~50% of the cells per field are displaced amacrine cells). For wt retinae, we can expect about 50-70% of those cells to be responsive, resulting in around 40 responsive RGCs per field. For *rd10*, this number is lower for ages greater than P30: e.g., for P45 we found ~25 responsive RGCs per field. If we further apply a confidence score threshold ($CS \geq 0.25$) to reliably identify cell types, this further reduces the number of identified RGCs per field. Given that we distinguished 32 RGC types, we can only identify up to 3 cells per field for most types, which does not allow for reliably identifying functional mosaics.

If the reviewer thinks that these considerations are helpful to the reader, we could add a short paragraph to the Methods or Discussion.

of genes causing RP is incorrect (references are 20+ years old & prior to completion of the Human Genome Project). Retnet says genes and loci for all types of RP are 88. Similarly, references for animal model diversity are 20-30 years old and are probably not the most reflective of where the field stands in 2024.

Thank you – we updated citations and numbers.

Are RGC types more vulnerable from their inherent cell biology or is upstream circuitry impacted, leading to type specific RGC changes? Please clarify text to reflect the changes are from impacted upstream changes. Unless that is not an accurate conclusion, in which case, please justify.

We revised the respective text.

It would add clarity for the reader to add in % changes in the text for figure 6.

Done (now Fig. 8).

Missing refs/discussion points:

It would be beneficial to include discussion on how these models/methods (below) differ and what that tells us about the system.

*Sekirnjak 2011 P23H rat MEA of RGCs: RF size decreased, RF time increased, ON had decreased firing rate.

*Yu 2017 S334ter-3 rat, MEA of RGCs: looked at cone pathway and found altered temporal RFs, impaired DS. Change in spatial RF was proportional to size of WT RF (bigger shrunk more). This paper is mentioned in text (ref 48) but hard to compare data due to the scaling in figure 6. Are there correlations to size or speed as seen by Yu et al.?

*Fransen et al. 2015 P23H rat: sharp decline in ON and ON-OFF RGCs, OFF preserved at P300.

*Stasheff et al. 2011 *rd1* and *rd10* (going to ignore *rd1*): ON and OFF impacted to a similar degree in *rd10*. More sustained responses (maybe).

We now discuss these findings in the revised manuscript.

Line 402, the last 'On' should be 'Off'

Fixed.

Missing information related to gain and signal-to-noise, which have both been implicated in prior studies about RGC responses during retinitis pigmentosa. Are there cell type specific changes to either? Please include.

The quality index (QI), which estimates how reliable a cell responds to a stimulus, can be used as a proxy for the signal-to-noise ratio (SNR). In Fig. 6A,B, we compared the QI distributions between the RGC types recorded in wt and *rd10* per age using the Jenson-Shannon divergence (JSD), which measures the similarity between distributions. A caveat of this analysis is that identifying cell types requires a good SNR (that is, above QI threshold). Nevertheless, we found that the JSDs were relatively stable for types belonging to the super-groups of ‘Uncertain’, ‘On-Off’, ‘Fast On’, and ‘Slow On’ RGCs. As responsive ‘Off’ RGC types mostly disappeared in *rd10* at a later age (Fig. 4D), we can assume their SNR to be strongly reduced. Moreover, since we see a general decrease in responsiveness in *rd10*, it seems likely that the overall SNR is decreasing with progressing degeneration.

The Results section of the revised manuscript now reflects these considerations.

REQUIRED ITEMS

- Author photo and profile. First or joint first authors are asked to provide a short biography (no more than 100 words for one author or 150 words in total for joint first authors) and a portrait photograph. These should be uploaded and clearly labelled together in a Word document with the revised version of the manuscript. See Information for Authors for further details.

Nadine Dyzskant earned her bachelor's degree in Biosciences from Westfälische-Wilhelms-University in Münster (Germany) in 2018, followed by a master's degree in biology from Carl-von-Ossietzky University in Oldenburg (Germany) in 2021. Her longstanding interest in neuroscience—particularly in the mechanisms underlying sensory systems—led her to the University of Tübingen, where she is currently pursuing a PhD in the laboratory of Prof. Thomas Euler at the Institute for Ophthalmic Research and the Centre for Integrative Neuroscience. Her research focuses on retinal function and the progression of retinal diseases, including conditions such as *Retinitis Pigmentosa*.

- You must start the Methods section with a paragraph headed Ethical approval (https://jp.msubmit.net/cgi-bin/main.plex?form_type=display_requirements#methods).

We have added this paragraph to the Methods.

Research must comply with The Journal's policies regarding animal experiments (<https://physoc.onlinelibrary.wiley.com/hub/animal-experiments>) and adherence to these policies must be stated in the manuscript.

We have added the respective statement and now comply with the Journal's policies.

Authors should confirm in their Methods section that their experiments were carried out according to the guidelines laid down by their institution's animal welfare committee, including an ethics approval reference number. The Methods section must contain a statement about access to food, water and housing, details of the anaesthetic regime: anaesthetic used, dose and route of administration, and method of killing the experimental animals.

We have completed this information and added the anaesthetic dose and route of administration.

- The reference list must be in alphabetical order, rather than numbered, to comply with our Journal format.

Done.

- Your manuscript must include a complete Additional Information section, including competing interests; funding; author contributions and acknowledgements.

We have added this section to the revised manuscript.

- Please upload separate high-quality figure files via the submission form.

- Please ensure that any tables are editable and in Word format, and wherever possible, embedded in the article file itself.

The tables are now editable and embedded in the Methods of the Word article file.

- Please ensure that the Article File you upload is a Word file.

We have converted the manuscript into the Word format.

- Your paper contains Supporting Information of a type that we no longer publish, including supplementary tables and figures. Any information essential to an understanding of the paper must be included as part of the main manuscript and figures. The only Supporting Information that we publish are video and audio, 3D structures, program codes and large data files. Your revised paper will be returned to you if it does not adhere to our Supporting Information Guidelines.

We reorganised the figures, for details see below.

- Papers must comply with the Statistics Policy: https://jp.msubmit.net/cgi-bin/main.plex?form_type=display_requirements#statistics.

The revised manuscript adheres to the Journal's Statistics Policy.

In summary:

- If $n \leq 30$, all data points must be plotted in the figure in a way that reveals their range and distribution. A bar graph with data points overlaid, a box and whisker plot or a violin plot (preferably with data points included) are acceptable formats.

- If $n > 30$, then the entire raw dataset must be made available either as supporting information, or hosted on a not-for-profit repository, e.g. FigShare, with access details provided in the manuscript.

- 'n' clearly defined (e.g. x cells from y slices in z animals) in the Methods. Authors should be mindful of pseudoreplication.

- All relevant 'n' values must be clearly stated in the main text, figures and tables.

- The most appropriate summary statistic (e.g. mean or median and standard deviation) must be used. Standard Error of the Mean (SEM) alone is not permitted.

- Exact p values must be stated. Authors must not use 'greater than' or 'less than'. Exact p values must be stated to three significant figures even when 'no statistical significance' is claimed.

- A Data Availability Statement is required for all papers reporting original data. This must be in the Additional Information section of the manuscript itself. It must have the paragraph heading 'Data Availability Statement'. All data supporting the results in the paper must be either: in the paper itself; uploaded as Supporting Information for Online Publication; or archived in an appropriate public repository. The statement needs to describe the availability or the absence of shared data. Authors must include in their statement: a link to the repository they have used, or a statement that it is available as Supporting Information; reference the data in the appropriate section(s) of their manuscript; and cite the data they have shared in the References section. Whenever possible, the scripts and other artefacts used to generate the analyses presented in the paper should also be publicly archived. If sharing data compromises ethical standards or legal requirements then authors are not expected to share it, but must note this in their statement. For more information, see our Statistics Policy.

A data availability statement was added to the revised manuscript.

- Please include an Abstract Figure file, as well as the Figure Legend text within the main article file. The Abstract Figure is a piece of artwork designed to give readers an immediate understanding of the research and should summarise the main conclusions. If possible, the image should be easily 'readable' from left to right or top to bottom. It should show the physiological relevance of the manuscript so readers can assess the importance and content of its findings. Abstract Figures should not merely recapitulate other figures in the manuscript. Please try to keep the diagram as simple as possible and without superfluous information that may distract from the main conclusion(s). Abstract Figures must be provided by authors no later than the revised manuscript stage and should be uploaded as a separate file during online submission labelled as File Type 'Abstract Figure'. Please also ensure that you include the figure legend in the main article file. All Abstract Figures should be created using BioRender. Authors should use The Journal's premium BioRender account to export high-resolution images. Details on how to use and access the premium account are included as part of this email.

A graphical summary was added to the manuscript.

Dear Dr Euler,

Re: JP-RP-2024-287643R1 "Photoreceptor degeneration has heterogeneous effects on functional retinal ganglion cell types" by Thomas Euler, Nadine Dyszkant, Jonathan Oesterle, Yongrong Qiu, Merle Harrer, Timm Schubert, and Dominic Gonschorek

We are pleased to tell you that your paper has been accepted for publication in The Journal of Physiology.

Please see Comments below for some minor amendments that can be made at proof stage.

Yours sincerely,

Nathan Schoppa
Senior Editor
The Journal of Physiology

If you would like to receive our 'Research Roundup', a monthly newsletter highlighting the cutting-edge research published in The Physiological Society's family of journals (The Journal of Physiology, Experimental Physiology, Physiological Reports, The Journal of Nutritional Physiology and The Journal of Precision Medicine: Health and Disease), please click this link, fill in your name and email address and select 'Research Roundup':
<https://www.physoc.org/journals-and-media/membernews>

- You can help your research get the attention it deserves! Check out Wiley's free Promotion Guide for best-practice recommendations for promoting your work at: www.wileyauthors.com/eeo/guide. You can learn more about Wiley Editing Services which offers professional video, design, and writing services to create shareable video abstracts, infographics, conference posters, lay summaries, and research news stories for your research at: www.wileyauthors.com/eeo/promotion.

EDITOR COMMENTS

Reviewing Editor:

The authors have satisfactorily addressed all major concerns of both reviewers. This study provides valuable insights into the physiological changes that accompany retinal degeneration.

Senior Editor:

Your revised manuscript has been reviewed by two expert reviewers and the Journal editors, and we are pleased to inform you that the manuscript has been accepted for publication. Reviewer 1 has raised a few very minor additional concerns that you should address, but this can be handled in the galley proof stage.

REFEREE COMMENTS

Referee #1:

I have no further comments. The authors have addressed all my concerns.

Referee #2:

The additional clarification regarding classification, noise, and light levels has improved the paper. I appreciate the authors' thoughtful responses and edits.

Small corrections:

- 1) Reference manager is throwing an error (pages 4, 8, 20, 24, 26).
- 2) I believe it would be useful to add average number of RGC responses per recording, as this aids the reader in comparing between studies. This could be added in the Methods section under Two-photon Ca²⁺ imaging.